# Anti-PD-1 cis-delivery of low-affinity IL-12 activates intratumoral CD8+T cells for systemic antitumor responses

Zhuangzhi Zou[1,2,5], Jiao Shen[1,2,5], Diyuan Xue[3,5], Hongjia Li[1,2,5], Longxin Xu[1,2,5], Weian Cao[3], Wenyan Wang[3], Yang-Xin Fu[3] ✉ & Hua Peng[1,4] ✉

Immune checkpoint blockade (ICB) therapies function by alleviating immunosuppression on tumor-infiltrating lymphocytes (TILs) but are often insufficient to fully reactivate these dysfunctional TILs. Although interleukin 12 (IL-12) has been used in combination with ICB to improve efficacy, this remains limited by severe toxicity associated with systemic administration of this cytokine. Here, we engineer a fusion protein composed of an anti-PD-1 antibody and a mouse low-affinity IL-12 mutant-2 (αPD1-mIL12mut2). Systemic administration of αPD1-mIL12mut2 displays robust antitumor activities with undetectable toxicity. Mechanistically, αPD1-mIL12mut2 preferentially activates tumor-infiltrating PD-1+CD8+T cells via high-affinity αPD-1 mediated *cis*-binding of low-affinity IL-12. Additionally, αPD1-mIL12mut2 treatment exerts an abscopal effect to suppress distal tumors, as well as metastasis. Collectively, αPD1-mIL12mut2 treatment induces robust systemic antitumor responses with reduced side effects.

Immunotherapies effectively treat tumors by enhancing the antitumor immune response. Immune checkpoint blockade (ICB), a prominent example of rapidly developing immunotherapies, has benefited many cancer patients. However, several challenges, such as a relatively low overall response rate[1,2] and acquired resistance[3], still impose restrictions on the widespread clinical application of ICB. Tumor-infiltrating lymphocytes (TILs) are susceptible to the immunosuppressive microenvironment and are driven into a dysfunctional state[4]. ICB is known to block inhibitory signals and partially alleviate immune suppression[5], although it is often not insufficient to fully reactivate these dysfunctional TILs.

Additional activation of TILs by cytokines enhances ICB efficacy and induces more potent antitumor immunity. Notably, IL-12, acting as a bridge between innate and adaptive immunity, plays a crucial role in promoting the differentiation of T helper (Th)−1 cells[6] and effector CD8+T cells[7]. IL-12 also enhances the effector function of activated T cells and natural killer (NK) cells, primarily by inducing interferon γ

(IFN-γ) production[8]. Importantly, IL-12 is indispensable for successful anti-PD-1 therapy[9]. Furthermore, resistance to checkpoint blockade in non−small cell lung cancer was overcome by a combination treatment of IL-2 and IL-12, which restores effector T-cell differentiation[10]. However, systemic application of IL-12 often results in severe toxicity in both preclinical and clinical trials[11,12]. Given the adverse effects that hinder clinical success of IL-12, numerous local delivery strategies have been proposed to mitigate side effects[13–15]. The primary objective of these approaches is to confine the bioactivity of IL-12 within tumor tissues, as systemic IL-12 administration frequently leads to the activation of peripheral effector cells and consequent toxicity.

One approach to enhance antitumor efficacy and reduce IL-12-related toxicity is to engineer pro-IL12 for tissue-specific activation within the tumor microenvironment (TME), which employ natural IL-12 receptors as masks to block IL-12 activity in the periphery[16,17]. As NK cells have been identified as the primary source of systemic IFN-γ in IL-12 treatment[17], leading to severe side effects, another approach is to

[1]Key Laboratory of Infection and Immunity, Institute of Biophysics, Chinese Academy of Sciences, 100101 Beijing, China. [2]University of Chinese Academy of Sciences, 100049 Beijing, China. [3]Department of Basic Medical Sciences, School of Medicine, Tsinghua University, 100084 Beijing, China. [4]Guangzhou Laboratory, Guangzhou 510320 Guangdong, China. [5]These authors contributed equally: Zhuangzhi Zou, Jiao Shen, Diyuan Xue, Hongjia Li, Longxin Xu. ✉ e-mail: Yangxinfu@tsinghua.edu.cn; hpeng@moon.ibp.ac.cn

utilize attenuated IL-12 mutants to preferentially activate CD8+T cells rather than NK cells, capitalizing on the differences in receptor expression[18]. A third strategy involves linking tumor-targeting elements, such as antibodies[19,20] or protein domains[21,22] with IL-12 to promote its retention in tumors.

Since PD-1 is expressed much more prominently within tumors than in the periphery[23,24], in the present study, we developed a low-affinity IL-12 mutant fused with a high-affinity anti-PD-1 antibody to enhance its specific binding to tumor-infiltrating PD-1+CD8+T cells rather than peripheral T or NK cells through *cis*-delivery. We demonstrate that this modified IL-12, guided by anti-PD-1 antibodies, ensures the preferential activation of PD-1+CD8+T cells locally in tumors, resulting in robust antitumor immune responses with minimal systemic toxicity.

## Results

### IL-12-mediated toxicity is attributed to NK cells

We observed that IL-12 prominently binds to NK cells but not splenic T cells in vitro (Supplementary Fig. 1a), suggesting that IL-12 might preferentially activate peripheral NK cells in our models. To explore whether NK cells are primarily responsible for IL-12-induced toxicity in our model system, NK cells were depleted by antibodies during systemic IL-12 treatment (Supplementary Fig. 1b). We measured the mouse body weight and collected the serum over different time points to characterize the side effects induced by IL-12. It was evident that IL-12 treatment led to severe weight loss. However, NK cell depletion completely abrogated this adverse effect (Fig. 1a). Cytometric bead arrays were used to determine serum cytokine levels. IL-12 treatment

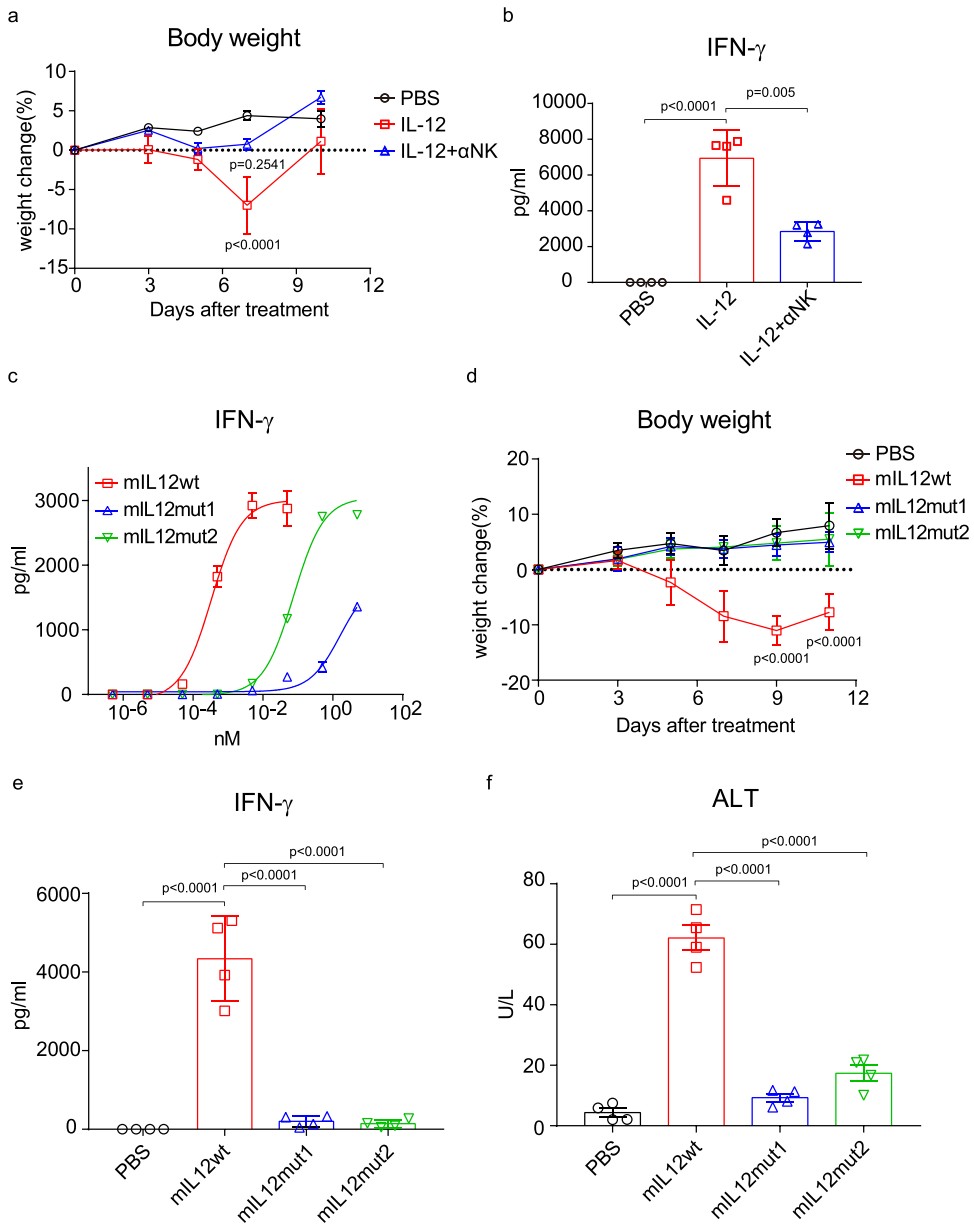

**Fig. 1 | Mutant IL-12 attenuates NK cell activation and toxicity. a, b** MC38 tumor-bearing mice (*n* = 4/group) were intraperitoneally treated with PBS or 5 μg IL-12 on days 14 and 17. For NK cell depletion, mice were intraperitoneally injected with 400 μg αNK1.1 antibody one day before treatment and then once every 3 days for three times. **a** The body weight change curve of mice after treatment. **b** The IFN-γ level in serum 6 h after the second treatment. **c** NK cells were incubated with mIL12wt, mIL12mut1, or mIL12mut2 for 48 h. The IFN-γ in the supernatant was

detected (*n* = 3). **d–f** MC38 tumor-bearing mice (*n* = 4/group) were intraperitoneally treated with PBS or 5 μg mIL12wt, mIL12mut1, or mIL12mut2 on days 14, 17, and 20. **d** The body weight change curve of mice after treatment. **e** The IFN-γ and **f** ALT in serum 24 h after the third treatment. All data are shown as mean ± SD from two to three independent experiments. The *P* value was determined by one-way (**b**, **e**, **f**) or two-way ANOVA (**a**, **d**).

markedly elevated IFN-γ serum levels, which were significantly reduced by NK cell depletion (Fig. 1b), consistent with the primary role of IFN-γ in mediating the side effects of IL-12 treatment[12]. Nevertheless, IL-12 treatment effectively suppressed MC38 tumor growth without NK cells (Supplementary Fig. 1c). These results suggest that IL-12-induced toxicity is attributed to NK cells, while its antitumor efficacy is largely independent of NK cells in our tumor model.

## Mutant IL-12 attenuates NK cell activation and toxicity

Several binding sites in the p19 subunit of IL-23 have been identified as essential for its effective binding to IL-23R[25]. Given the structural similarities of p19 (in IL-23) and p35 (in IL-12), we speculated that some sites in p35 might also be crucial for high binding affinity with receptors and, subsequently, high bioactivity on effector cells. To reduce the severe toxicity caused by IL-12 in vivo, IL-12 bioactivity, particularly on NK cells, needed to be suppressed. The bioactivity of several IL-12 mutants constructed by single-site substitution was measured in vitro using the HEK-IL-12 reporter cell line. Either the Y163A or K166A mutation resulted in an approximately 10-fold reduction in bioactivity (Supplementary Fig. 1d). Similarly, the Y163R mutation reduced IL-12 bioactivity in engineered Ba/F3 cells[26]. Moreover, the Y163A/K166A double site mutation further attenuated bioactivity by approximately 100-fold (Supplementary Fig. 1e). In addition to p35, several amino acid sites in p40 have been reported to be essential for functional IL-12 signaling[18]. Therefore, we compared the effect of p35 or p40 mutation on IL-12 bioactivity. In addition to the p35 Y163/K166A double site mutant (mIL12mut1), we also constructed a p40 E81A/F82A double site mutant (mIL12mut2). Compared with wild-type IL-12 (mIL12wt), both mutants induced notably weaker p-STAT4 signaling and decreased IFN-γ production in NK cells (Fig. 1c; Supplementary Fig. 1f), confirming the extremely attenuated bioactivities of these mutants. Moreover, the substantially reduced bioactivities of these mutants were also observed in activated CD8+T cells in vitro (Supplementary Fig. 1g, h).

To explore whether these mutants can reduce side effects in vivo, MC38 tumor-bearing mice were treated with WT or mutated IL-12. The mIL12wt treatment caused severe weight loss, but this adverse effect was not observed in mice treated with either mIL12mut1 or mIL12mut2 (Fig. 1d). Moreover, neither mIL12mut1 nor mIL12mut2 elevated the level of IFN-γ in contrast to the high concentration in serum induced by mIL12wt (Fig. 1e). IL-12 treatment can cause liver damage characterized with elevated transaminases in serum[12]. We also found a significantly elevated alanine aminotransferase (ALT) serum level after mIL12wt treatment compared to a negligible increase in mIL12mut1 or mIL12mut2-treated mice (Fig. 1f). Collectively, both mIL12mut1 and mIL12mut2 can effectively reduce IL-12-related toxicity by greatly attenuating IL-12 bioactivity on NK cells.

## Anti-PD1-mIL12mut2 preferentially binds to tumor-infiltrating PD-1+CD8+T cells

Persistent antigen stimulation in tumors promoted much higher levels of PD-1 expression on TILs than on peripheral T cells[27]. Moreover, high PD-1 expression was also identified in tumor-reactive CD8+T cells in tumors[28]. We found that CD8+T cells in tumors exclusively expressed a higher level of PD-1 in the MC38 tumor model (Fig. 2a). NK cells from different peripheral tissues expressed a markedly lower level of PD-1 than intratumoral CD8+T cells (Supplementary Figs. 2a, 6a). Furthermore, in tumors, CD8+T cells showed a significantly higher level of PD-1 expression than CD4+T cells or NK cells (Fig. 2b; Supplementary Figs. 2b, 6b). Therefore, we hypothesized that anti-PD-1 antibodies would be capable of effectively delivering IL-12 mutants to tumor-infiltrating PD-1+CD8+T cells instead of peripheral NK or T cells for greater antitumor efficacy while circumventing NK-mediated toxicity.

When the binding affinity of these IL-12 mutants was measured, although mIL12mut1 showed substantially decreased binding affinity with the IL-12Rβ2 protein alone (Supplementary Fig. 2c), we found that mIL12mut1 maintained effective binding on HEK cells in vitro (Fig. 2c). Moreover, the binding of mIL12mut2 on HEK cells was severely impaired (Fig. 2c). Additionally, we obtained similar results in activated T cells in vitro (Supplementary Fig. 2d). To avoid unnecessary binding on peripheral effector cells, which may also reduce PD-1-mediated tumor targeting effect, we first tested whether mIL12mut2 with lower affinity was preferable for fusion with αPD-1 antibodies. Knobs-into-holes technology was used to construct this asymmetric fusion protein, linking one molecule of mIL12mut2 at the C terminus of the Fc domain of anti-PD-1 antibodies, named αPD1-mIL12mut2 (Fig. 2d). The fusion protein included whole anti-PD-1 antibody for optimal binding affinity and one molecule of mIL12mut2 to increase the target-effector ratio. Meanwhile, αEGFR-mIL12mut2 with an anti-human EGFR antibody was constructed as a nontargeting control fusion protein. When cell suspension from MC38 tumor tissue was incubated with these fusion proteins in vitro, αPD1-mIL12mut2 preferentially bound to CD8+T cells rather than CD4+T cells or NK cells. Moreover, αPD1-mIL12mut2 showed an enhanced binding ability to CD8+T cells compared to αEGFR-mIL12mut2 (Fig. 2e), consistent with the high PD-1 expression on intratumoral CD8+T cells (Fig. 2b). These results indicated that αPD-1 antibodies play a dominant role in mediating the binding of αPD1-mIL12mut2 to intratumoral CD8+T cells. Moreover, αPD1-mIL12mut2 showed weaker binding on peripheral effector cells in sharp contrast to intratumoral CD8+T cells (Supplementary Fig. 2e).

Next, we explored whether αPD1-mIL12mut2 could target tumors in vivo. The fusion protein αPD1-mIL12mut2 was intraperitoneally injected into tumor-bearing mice, and different tissues were collected to measure the concentration of the fusion protein. The concentration of αPD1-mIL12mut2 was significantly higher in the tumor tissue than in other tissues (Fig. 2f), indicating that αPD1-mIL12mut2 can effectively target the tumor. Thus, we constructed a fusion protein αPD1-mIL12mut2 that preferentially accumulates in tumors and binds to intratumoral CD8+T cells.

## Anti-PD1-mIL12mut2 acquires enhanced bioactivity via αPD-1-mediated cis-binding

Since PD-1 and IL-12 receptors are both highly expressed on intratumoral CD8+T cells, especially tumor-reactive T cells, it is worth exploring whether αPD-1 antibodies would help to restore the attenuated bioactivity of IL-12 on effector cells. HEK cells were engineered to express mouse PD-1 (HEK-mPD-1 cells). First, we found that the binding of αPD1-mIL12mut2 to HEK-mPD-1 cells was more robust than that of αEGFR-mIL12mut2 (Fig. 3a), again confirming that αPD-1 antibodies, not IL-12, dominated the binding of αPD1-mIL12mut2 to PD-1 positive effector cells. Next, we investigated whether this enhanced binding ability could contribute to the restored bioactivity of αPD1-mIL12mut2. Without PD-1-mediated binding, αPD1-mIL12mut2 showed the same bioactivity as αEGFR-mIL12mut2 on HEK cells (Supplementary Fig. 3a). However, the bioactivity of αPD1-mIL12mut2 was much more potent than that of αEGFR-mIL12mut2 on HEK-mPD-1 cells (Fig. 3b). Furthermore, when HEK-mPD-1 cells were blocked by αPD-1 antibodies, the bioactivity of αPD1-mIL12mut2 was reduced to a low level, as induced by αEGFR-mIL12mut2 (Fig. 3b), suggesting that PD-1 binding was the prerequisite for enhanced IL-12 bioactivity on PD-1 positive cells. Then, HEK-mPD-1 cells were blocked with serially diluted αPD-1 antibodies to mimic different PD-1 expression levels on HEK cells. As the concentration of blockade antibody increased, the bioactivity of αPD1-mIL12mut2 gradually decreased (Fig. 3c), further confirming that a high level of PD-1 expression on effector cells was essential for efficacious IL-12 signaling activated by αPD1-mIL12mut2. These results demonstrated that αPD-1 antibodies profoundly facilitated the binding of IL-12 and subsequently resulted in enhanced bioactivity of IL-12 on PD-1 positive effector cells, possibly through cis-delivery. Moreover, we detected αPD1-mIL12mut1 bioactivity using the same method. The αPD1-mIL12mut1 showed similar bioactivities to

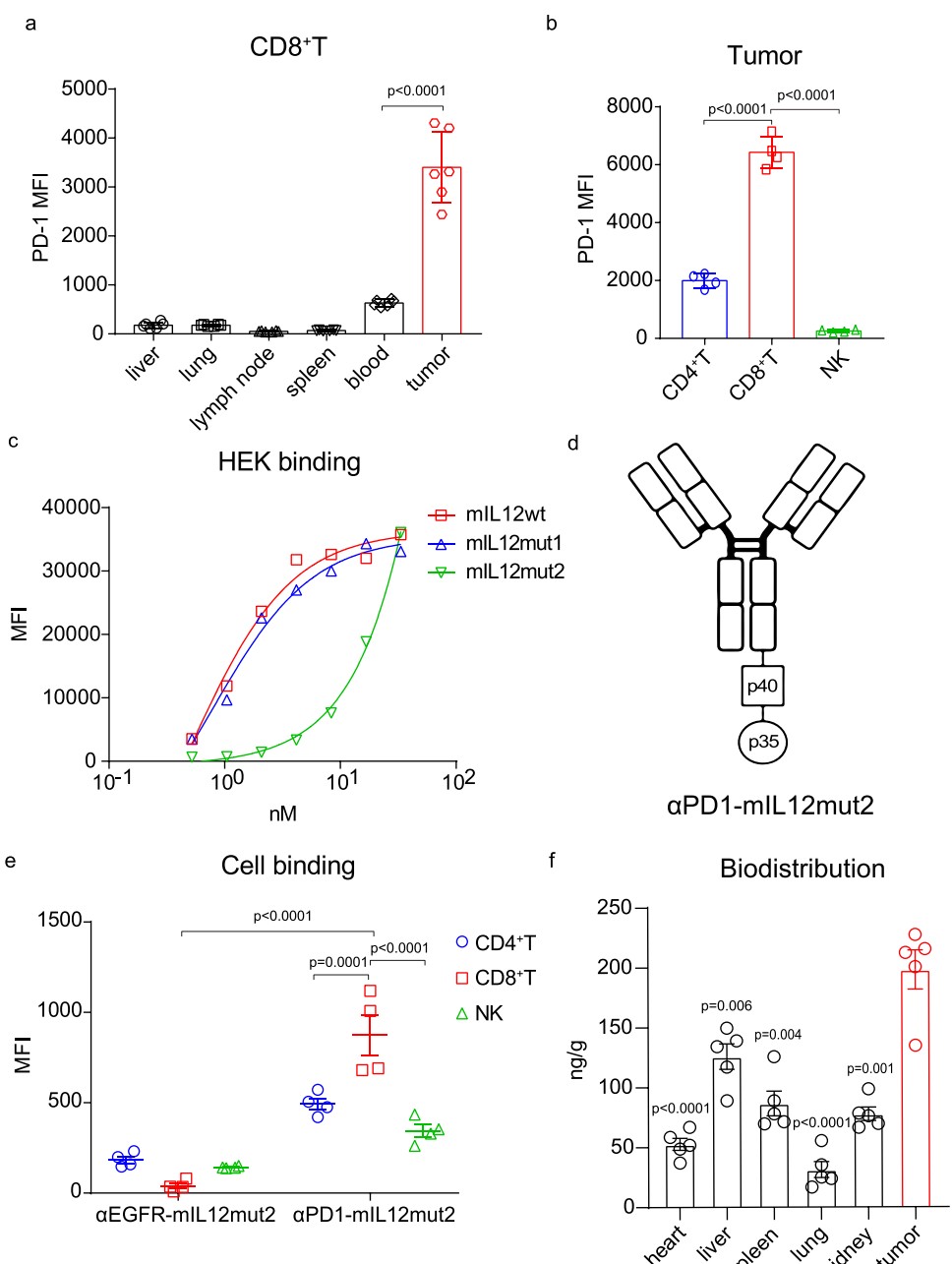

**Fig. 2 | Anti-PD1-mIL12mut2 preferentially binds to tumor-infiltrating PD-1+CD8+T cells. a** The PD-1 expression on CD8+T cells in different tissues from MC38 tumor-bearing mice ($n = 6$ mice). **b** The PD-1 expression on CD4+T cells, CD8+T cells, or NK cells in MC38 tumor ($n = 4$ mice). **c** HEK cells were incubated with mIL12wt, mIL12mut1, or mIL12mut2 in vitro. Protein binding to HEK cells was detected by flow cytometric analysis ($n = 2$). **d** Schematic diagram of αPD1-mIL12mut2. **e** Single-cell suspension of MC38 tumor tissue ($n = 4$ mice) was incubated with αEGFR-mIL12mut2 or αPD1-mIL12mut2 in vitro. Protein binding to CD4+T cells, CD8+T cells, or NK cells was detected by flow cytometric analysis. **f** MC38 tumor-bearing mice ($n = 5$) were intraperitoneally injected with 20 μg αPD1-mIL12mut2 and sacrificed 24 h after treatment. Mice tissues were collected and homogenized. The concentration of fusion protein in the supernatant was measured and normalized using ELISA. MFI mean fluorescence intensity. All data are shown as mean ± SD from two to three independent experiments. The *P* value was determined by one-way (**a**, **b**), two-way ANOVA (**e**) or unpaired *t* test (**f**).

those of αEGFR-mIL12mut1 in both HEK (Supplementary Fig. 3b) and HEK-mPD-1 cells (Supplementary Fig. 3c), suggesting that αPD-1 antibodies failed to restore the bioactivity of mIL12mut1. To examine the expression levels of PD-1 and IL-12 receptors on reporter cells and T cells derived from tumor tissues, we quantify the receptor levels on the plasma membrane using Quantitative Flow Cytometry[29]. To establish a standard curve, the kit containing beads conjugated with four distinct standard levels of PE fluorescence was utilized (Supplementary Fig. 5a–c). Subsequently, the PD-1 and IL12R counts on HEK-mPD-1 reporter cells and intratumoral CD8+T cells, respectively, were

determined using PE-conjugated antibodies (Supplementary Fig. 5d–g). While reporter cells exhibited significantly higher expression levels of PD-1 and IL-12 receptors compared to intratumoral CD8+T cells, the ratios of PD-1 to IL-12 receptor were comparable between these cell types, suggesting that this genetically modified HEK cell line can be used to monitor the activity of αPD1-mIL12, most likely on T cells.

To test whether αPD-1 can directly target CD8+T cells via *cis*-binding, we first compared the activation of WT or PD-1 KO (knockout) CD8+T cells treated with αPD1-mIL12mut2 or αEGFR-mIL12mut2

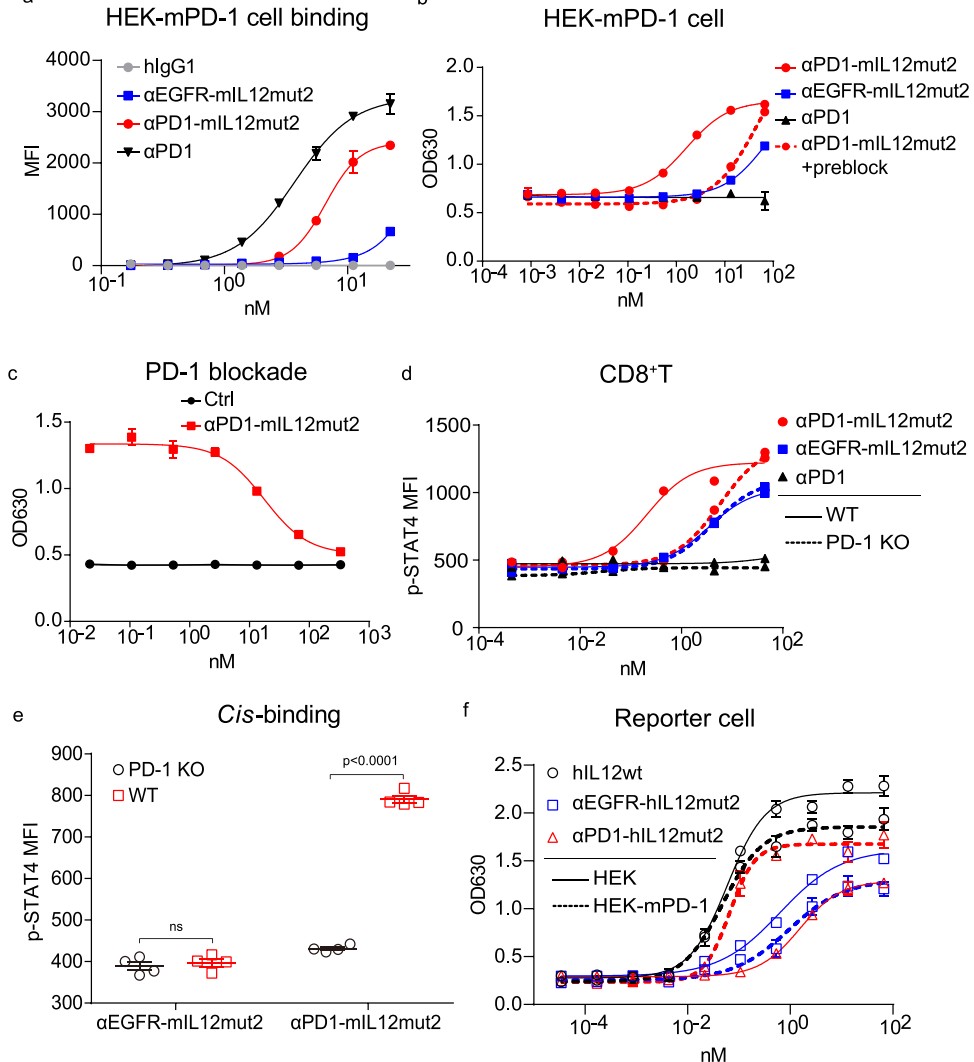

**Fig. 3 | Anti-PD1-mIL12mut2 acquires enhanced bioactivity via αPD-1-mediated *cis*-binding. a** HEK-mPD-1 cells were incubated with serially diluted hIgG1, αEGFR-mIL12mut2, αPD1-mIL12mut2, or anti-PD-1 antibody in vitro. Protein binding to HEK-mPD-1 cells was detected by flow cytometric analysis (*n* = 4 replicates). **b** The activity of αEGFR-mIL12mut2 or αPD1-mIL12mut2 was detected using HEK-mPD-1 cells. For PD-1 blockade, 20 μg/ml anti-PD-1 antibody was added into the incubation system of HEK-mPD-1 cells and αPD1-mIL12mut2 (*n* = 3 replicates). **c** Different concentration of anti-PD-1 antibody was added into the incubation system of HEK-mPD-1 cells and αPD1-IL12mut2. The activity of αPD1-mIL12mut2 was detected (*n* = 3 replicates). Cell culture medium without fusion protein was used to incubate HEK-mPD-1 cells as negative control. **d** Pre-activated WT or PD-1 KO CD8+T cells

were incubated with αEGFR-mIL12mut2 or αPD1-mIL12mut2 for 0.5 h in vitro. The p-STAT4 in CD8+T cells was detected by flow cytometric analysis (*n* = 2). **e** Pre-activated WT CD8+T cells were labeled with CTV. An equal number of CTV-labeled WT CD8+T cells and non-labeled PD-1 KO CD8+T cells were mixed and then incubated with αEGFR-mIL12mut2 or αPD1-mIL12mut2 for 0.5 h in vitro. The p-STAT4 in CD8+T cells was detected by flow cytometric analysis (*n* = 4 mice/group). MFI, mean fluorescence intensity. **f** The activity of hIL12wt, αEGFR-hIL12mut2, or αPD1-hIL12mut2 was examined using the HEK or HEK-mPD-1 cells (*n* = 3 replicates). All data are shown as mean ± SD from two to three independent experiments. The P value was determined by two-way ANOVA (**e**).

(Supplementary Fig. 3d). We observed that both fusion proteins induced comparable but weak p-STAT4 signaling in PD-1 KO CD8+T cells. However, αPD1-mIL12mut2 induced significantly stronger p-STAT4 signaling than αEGFR-mIL12mut2 in WT CD8+T cells (Fig. 3d), suggesting that αPD-1-mediated binding successfully restored the bioactivity of IL-12 on PD-1+CD8+T cells. Next, we wanted to explore whether αPD-1 antibodies might bring the IL-12 mutant to PD-1+CD8+T cells via *cis*-binding. WT CD8+T cells were labeled with Cell Trace Violet (CTV) and then mixed with equinumerous unlabeled PD-1 KO CD8+T cells. In the mixture, αEGFR-mIL12mut2 induced the same and low level of p-STAT4 in both cells. In contrast, αPD1-mIL12mut2 potently induced increased level of p-STAT4 in WT CD8+T cells but not in PD-1 KO CD8+T cells (Fig. 3e). This result confirmed that αPD1-mIL12mut2 preferred to activate PD-1+CD8+T cells in a *cis*-binding manner.

To explore whether αPD-1 antibody-mediated *cis*-binding could restore the bioactivity of human IL-12 mutants, we also constructed human IL-12 mutant-based fusion proteins. When incubated with HEK-mPD-1 cells, αEGFR-hIL12mut2 displayed much weaker binding than αEGFR-hIL12mut1 (Supplementary Fig. 3e), indicating that human IL12mut2, not mut1, caused significantly reduced binding affinity. Meanwhile, αPD1-hIL12mut2 had extensively restored binding affinity to HEK-mPD-1 cells, similar to that of αPD1-hIL12mut1 (Supplementary Fig. 3e). Consequently, only αPD1-hIL12mut2 showed potent enhanced bioactivity merely on HEK-mPD-1 cells (Fig. 3f), while αPD1-hIL12mut1 still showed impaired bioactivity in both HEK and HEK-mPD-1 cells (Supplementary Fig. 3f). These results demonstrate that αPD-1-mediated *cis*-binding can also effectively restore the bioactivity of human p40-mutated IL-12.

### Anti-PD1-mIL12mut2 efficiently suppresses tumor growth without toxicity

Since αPD1-mIL12mut2 could target tumors with enhanced bioactivity on PD-1[+]CD8[+]T cells, we considered whether αPD1-mIL12mut2 could effectively suppress tumor growth in vivo. MC38 tumor cells were subcutaneously inoculated on the back of mice, and well-established tumors were treated beginning on day 14. As the dose increased, αPD1-mIL12mut2 treatment elicited an increasingly enhanced suppressive

effect on tumor growth (Fig. 4a), suggesting that αPD1-mIL12mut2 treatment delayed tumor growth in a dose-dependent manner. Either αPD-1 or αEGFR-mIL12mut2 alone or in combination showed limited inhibition of tumor growth compared to αPD1-mIL12mut2 (Fig. 4b), indicating that αPD1-mIL12mut2 had superior antitumor efficacy than the nontargeting IL-12 mutant. Next, the MC38-EGFR5 tumor model was used to investigate the importance of *cis*-delivering in vivo. MC38 was engineered to express mutated mouse EGFR, which can be

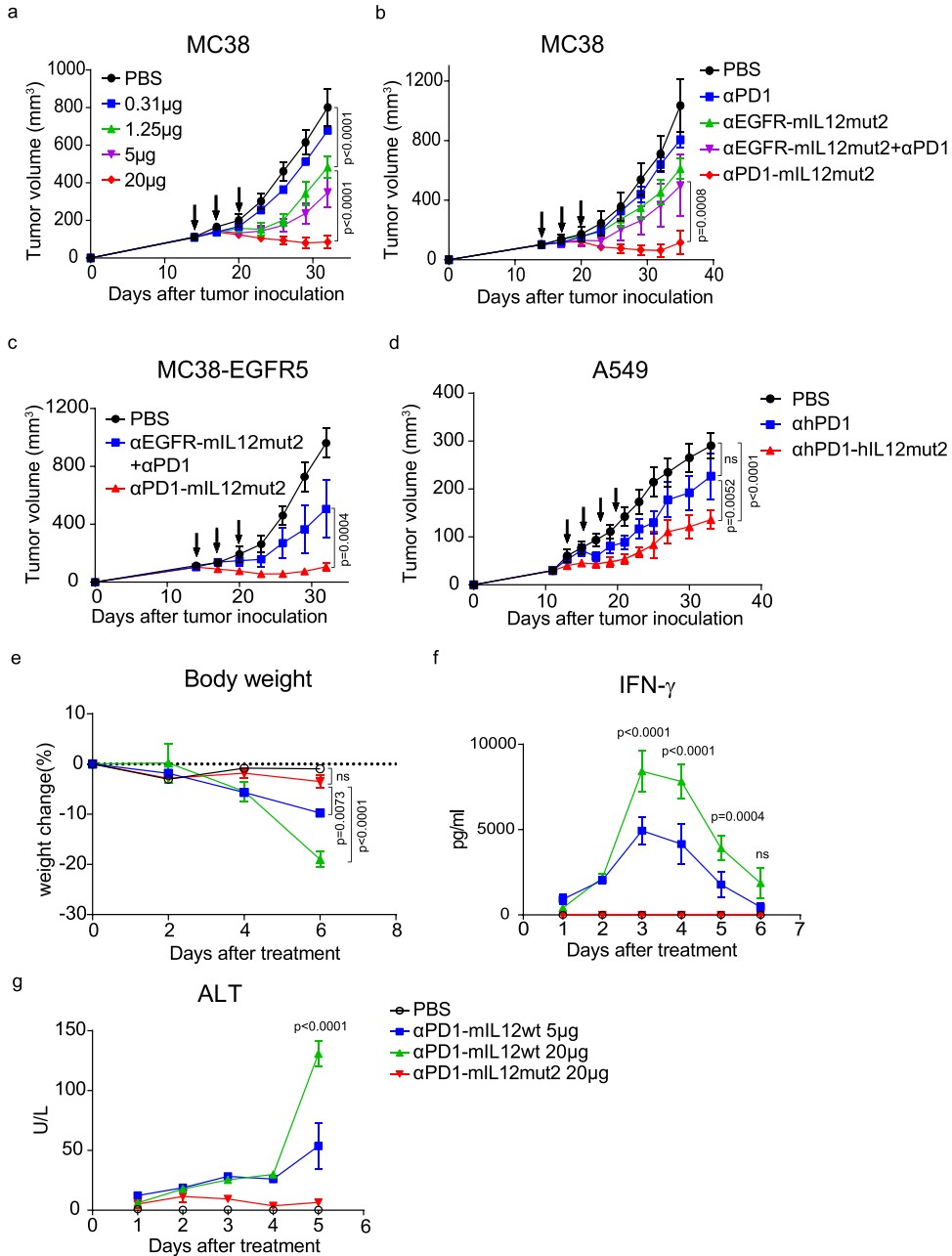

**Fig. 4 | Anti-PD1-mIL12mut2 effectively suppresses tumor growth without toxicity. a** C57BL/6 mice were inoculated with 5 ×10[5] MC38 tumor cells. Tumor-bearing mice (*n* = 5/group) were intraperitoneally treated with PBS or different doses of αPD1-mIL12mut2 on days 14, 17 and 20. The tumor growth of mice was recorded. **b** MC38 tumor-bearing mice (*n* = 4/group) were intraperitoneally treated with PBS or equimolar αPD-1 or αEGFR-mIL12mut2 or the mixture of αPD-1 and αEGFR-mIL12mut2 or αPD1-mIL12mut2 on days 14, 17 and 20. The tumor growth of mice was measured. **c** C57BL/6 mice were inoculated with 5 ×10[5] MC38-EGFR5 tumor cells. Tumor-bearing mice (*n* = 4/group) were intraperitoneally treated with PBS or an equimolar mixture of αPD-1 and αEGFR-mIL12mut2 or αPD1-mIL12mut2

on days 14, 17, and 20. The tumor growth of mice was monitored. **d** Humanized mice were inoculated with 2 ×10[6] A549 tumor cells. Tumor-bearing mice (*n* = 4/group) were intraperitoneally treated with PBS or equimolar αhPD-1 or αhPD1-hIL12mut2 on days 12, 15, 18 and 21. The tumor growth of mice was measured. **e**–**g** MC38 tumor-bearing mice (*n* = 3/group) were intraperitoneally treated once with PBS or 5 µg αPD1-mIL12wt or 20 µg αPD1-mIL12wt or 20 µg αPD1-mIL12mut2. **e** The body weight change curve of mice. **f** IFN-γ and **g** ALT in serum were measured after treatment. All data are shown as mean ± SEM from two to three independent experiments. The *P* value was determined by two-way ANOVA (**a**–**g**).

recognized and bound by anti-human EGFR antibodies. In the MC38-EGFR5 tumor model, αPD1-mIL12mut2 treatment also suppressed tumor growth more effectively than the combination therapy of αEGFR-mIL12mut2 and αPD-1 (Fig. 4c). To ascertain whether the diminished antitumor activity of αEGFR-mIL12mut2 stemmed from weaker tumor binding, we quantified the number of EGFR on MC38-EGFR5 tumor cells (Supplementary Fig. 5h), revealing a significantly higher count compared to PD-1 on intratumoral CD8+T cells (Supplementary Fig. 5f). Additionally, we analyzed the fusion protein binding efficacy to their respective target cells by incubating tumor tissue suspensions with each fusion protein. Our observation indicated that αPD1-mIL12mut2 primarily bound to CD8+T cells rather than tumor cells, whereas αEGFR-mIL12mut2 exhibited a stronger affinity to tumor cells over CD8+T cells. Notably, the binding kinetics of αEGFR-mIL12mut2 to tumor cells surpassed that of αPD1-mIL12mut2 to CD8+T cells (Supplementary Fig. 5i). This outcome underscored that the reduced antitumor efficacy of αEGFR-mIL12mut2 was attributable to its inability to recruit IL-12 mutant bioactivity to the intratumoral T cells. Altogether, these results suggested that CD8+T cell-targeting fusion protein αPD1-mIL12mut2 had superior antitumor efficacy to the tumor cell-targeting fusion protein αEGFR-IL12mut2, demonstrating that PD-1 targeting is a practical design to maximize the antitumor activity of the IL-12 mutant.

To investigate the antitumor activity of the human version of αhPD1-hIL12mut2 in vivo, we transferred human PBMCs into NOD/SCID/IL2rγ$^{null}$ (NSG) mice. Then, these mice were inoculated with A549 human lung carcinoma cells. Anti-human PD-1 antibody treatment alone showed negligible effect on tumor growth. In contrast, αhPD1-hIL12mut2 treatment significantly suppressed tumor growth (Fig. 4d), suggesting that IL-12 enhances the antitumor efficacy of anti-PD-1 antibodies in the form of a T-cell-targeting fusion protein.

To further validate the safety of αPD1-mIL12mut2 when used in vivo, tumor-bearing mice were intraperitoneally injected with αPD1-mIL12wt or αPD1-mIL12mut2. We observed that αPD1-mIL12wt resulted in dose-dependent weight loss, whereas αPD1-mIL12mut2 did not cause any weight change (Fig. 4e). The serum levels of inflammatory cytokines, such as IFN-γ and TNF-α quickly increased after αPD1-mIL12wt administration. Nevertheless, αPD1-mIL12mut2 was well-tolerated and prevented the serum level increases of these cytokines (Fig. 4f; Supplementary Fig. 3g). Moreover, αPD1-mIL12wt resulted in a notably higher level of ALT and AST than αPD1-mIL12mut2 in serum (Fig. 4g; Supplementary Fig. 3h). Altogether, αPD1-mIL12mut2 acquired a wider antitumor therapeutic window.

### Anti-PD1-mIL12mut2-induced antitumor effect depends on intratumoral CD8+T cells and IFN-γ

Although we confirmed that NK cells were dispensable for the IL-12-induced antitumor effect in our tumor models, how αPD1-mIL12mut2 elicits robust antitumor immune responses remains unclear. To clarify this mechanism, we first tested the antitumor efficacy of αPD1-mIL12mut2 in *Rag1*$^{-/-}$ mice lacking adaptive immune cells. The antitumor efficacy of αPD1-mIL12mut2 was completely abrogated in *Rag1*$^{-/-}$ mice (Fig. 5a), indicating that adaptive immunity was essential for αPD1-mIL12mut2 mediated antitumor activity. T cells, including CD8+T and Th1 cells, are known to respond to IL-12 stimulation. Therefore, CD4+T or CD8+T cells or both cell types were depleted to explore the role of these cells in the antitumor activity of αPD1-mIL12mut2. CD8+T cell depletion alone or together with CD4+T cell depletion entirely abolished the antitumor effect of αPD1-mIL12mut2 (Fig. 5b), demonstrating that CD8+T cells were indispensable for αPD1-mIL12mut2-induced robust antitumor responses. Interestingly, CD4+T cell depletion alone during αPD1-mIL12mut2 treatment enhanced antitumor efficacy (Fig. 5b). All CD4+T cells including T$_{reg}$ cells were depleted by anti-CD4 antibodies, and T$_{reg}$ cells, known as immunosuppressive cells, may restrain αPD1-mIL12mut2 induced antitumor responses. After

αPD1-mIL12mut2 treatment, the frequency and number of T$_{reg}$ cells significantly decreased (Supplementary Fig. 4a, b), and CD8+T cells to T$_{reg}$ ratio was increased (Supplementary Fig. 4c). These results suggested that T$_{reg}$ cells were negatively correlated with the outcome of αPD1-mIL12mut2 therapy, further implying a dominant role of T$_{reg}$ cells in CD4+T cells to inhibit the antitumor immune responses in this tumor model. IFN-γ is the major downstream effector of IL-12 and plays a vital role in antitumor immune responses[30]. After αPD1-mIL12mut2 treatment, both the frequency and number of IFN-γ+CD8+T cells in tumors were significantly increased (Fig. 5c; Supplementary Figs. 4d, 6c), confirming that αPD1-mIL12mut2 could potently stimulate intratumoral CD8+T cells to produce more IFN-γ. Moreover, αPD1-mIL12mut2 treatment also stimulated intratumoral CD8+T cells to produce more granzyme B (Fig. 5d; Supplementary Figs. 4e, 6d). Indeed, IFN-γ signaling was especially essential for the antitumor efficacy of αPD1-mIL12mut2, which was completely abolished when IFN-γ was neutralized by antibodies (Fig. 5e). Next, to explore whether pre-existing intratumoral T cells were sufficient for the αPD1-mIL12mut2-induced antitumor effect, FTY720 was used to prevent new T-cell egress from the lymph node and FTY720 blockade diminished T cells in the periphery (Supplementary Fig. 4f). However, the antitumor efficacy of αPD1-mIL12mut2 was unaffected (Fig. 5f), suggesting that pre-existing intratumoral T cells were sufficient to mediate the antitumor effect of αPD1-mIL12mut2. Thus, αPD1-mIL12mut2 suppressed tumor growth through intratumoral CD8+T cell activation and IFN-γ production.

### Anti-PD1-mIL12mut2 preferentially activates PD-1+Tim3+CD8+T cells in tumors

Tumor-infiltrating CD8+T cells are susceptible to exhaustion in the suppressive tumor microenvironment, characterized with the upregulated expression of multiple coinhibitory receptors, such as PD-1 and TIM-3. To explore the effect of αPD1-mIL12mut2 treatment on TILs, TILs were grouped into three subsets: PD-1-, PD-1+TIM-3-, and PD-1+TIM-3+ cells according to the expression of PD-1 and TIM-3 (Supplementary Fig. 6e). Among these subsets, PD-1+TIM-3+CD8+T cells showed a higher level of PD-1 expression than PD-1+TIM-3-CD8+T cells (Supplementary Fig. 4g), consistent with the widely known role of PD-1+TIM-3+ cells as terminal exhausted cells in tumors. It has been reported that IL-12 induced Tim-3 expression and caused T-cell exhaustion in B-cell lymphoma[31]. In the MC38 tumor model, the frequency of these cell subsets barely changed after αPD1-mIL12mut2 treatment (Supplementary Fig. 4h), suggesting that αPD1-mIL12mut2 does not promote T-cell exhaustion. Furthermore, the proportion of IFN-γ+CD8+T cells was significantly increased only in PD-1+TIM-3+CD8+T cells (Supplementary Fig. 4i). This result demonstrated that αPD1-mIL12mut2 could enhance the effector function of PD-1+TIM-3+CD8+T cells in tumors.

### Anti-PD1-mIL12mut2 treatment induces systemic antitumor immune responses

Tumor metastasis is the primary cause of clinical treatment failure and cancer-related life-threatening diseases. It is essential to induce robust systemic antitumor immune responses to control distal tumors and suppress metastasis during tumor immunotherapy. To achieve this, we first used the double-flank MC38 tumor model to investigate whether local αPD1-mIL12mut2 treatment can induce an abscopal effect. MC38 tumor cells were subcutaneously injected into the right flank, and two days later, MC38 tumor cells were subcutaneously injected into the left flank. The tumor on the right flank was intratumorally injected with αPD1-mIL12mut2 (Fig. 6a). Local αPD1-mIL12mut2 treatment effectively suppressed tumor growth in both the right and left flanks (Fig. 6b, c). FTY720 blockade did not affect the antitumor efficacy in the right flank (Fig. 6b). However, it completely abrogated the tumor control effect in the left flank (Fig. 6c). These results indicated that

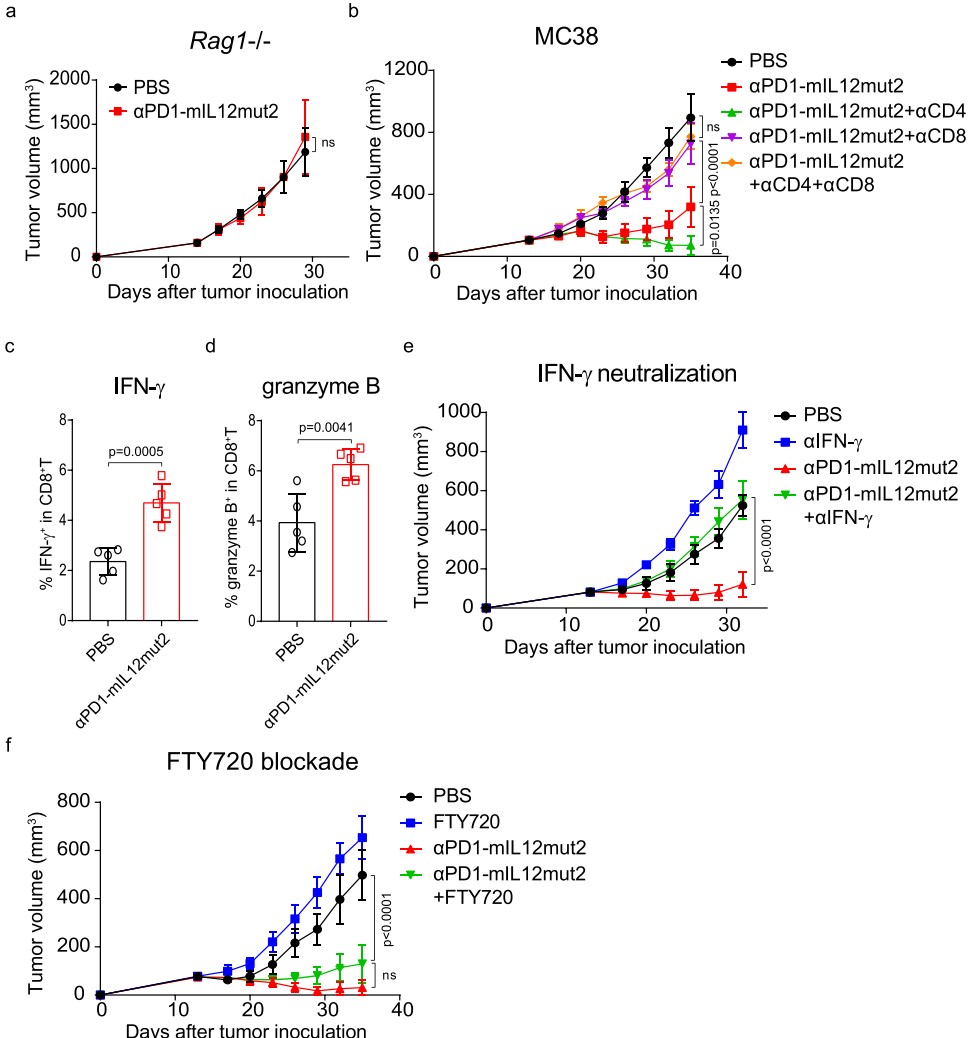

**Fig. 5 | Anti-PD1-mIL12mut2 induced antitumor effect depends on intratumoral CD8⁺T cells and IFN-γ. a** *Rag1⁻/⁻* mice were inoculated with 5 ×10⁵ MC38 tumor cells. Tumor-bearing mice (*n* = 5/group) were intraperitoneally treated with PBS or 10 μg αPD1-mIL12mut2 on days 14, 17, and 20. The mouse tumor growth curve was recorded. **b** C57BL/6 mice were inoculated with 5 ×10⁵ MC38 cells. Tumor-bearing mice (*n* = 5/group) were intraperitoneally treated with PBS or 20 μg αPD1-mIL12mut2 on days 14, 17 and, 20. For CD4⁺T or CD8⁺T cell depletion, mice were intraperitoneally injected with 200 μg αCD4 or αCD8 antibody one day before treatment and then once every 3 days for three times. The tumor growth curve of mice was monitored. **c** The frequency of IFN-γ⁺CD8⁺T cells in the tumors after αPD1-mIL12mut2 treatment (*n* = 5 mice/group). **d** The frequency of granzyme B⁺CD8⁺T cells in tumor after αPD1-mIL12mut2 treatment (*n* = 5 mice/group). **e** C57BL/6 mice were inoculated with 5 ×10⁵ MC38 tumor cells. Tumor-bearing mice

(*n* = 5/group) were intraperitoneally treated with PBS or 20 μg αPD1-mIL12mut2 on days 14, 17, and 20. For IFN-γ neutralization, mice were intraperitoneally injected with 500 μg αIFN-γ antibody one day before treatment and then once every 3 days for three times. The tumor growth curve of mice was measured. **f** C57BL/6 mice were inoculated with 5 ×10⁵ MC38 tumor cells. Tumor-bearing mice (*n* = 4/group) were intraperitoneally treated with PBS or 20 μg αPD1-mIL12mut2 on days 14, 17, and 20. Mice were intraperitoneally injected with 25 μg FTY720 one day before treatment and then once every other day for five times to block T cells exiting from the lymph node. The tumor growth curve of mice was monitored. Data are shown as mean ± SD (**c, d**), or mean ± SEM (**a, b, e, f**) from two to three independent experiments. The *P* value was determined by unpaired *t* test (**c, d**) or two-way ANOVA (**a, b, e, f**).

αPD1-mIL12mut2 treatment could control primary tumors through TILs and distal tumors through T cells egressing from the lymph node.

The experimental pulmonary metastasis model was used to explore whether αPD1-mIL12mut2 treatment can suppress tumor metastasis. B16F10 tumor cells were subcutaneously inoculated on the back of mice, and other B16F10 tumor cells were intravenously injected into mice one day before αPD1-mIL12mut2 treatment (Fig. 6d). Intratumoral αPD1-mIL12mut2 treatment significantly delayed tumor growth in situ (Fig. 6e) and suppressed tumor metastasis in the lung (Fig. 6f). Collectively, αPD1-mIL12mut2 treatment can induce systemic antitumor responses to clear distal tumors and suppress metastasis.

In addition, when the mice with complete MC38 tumor regression after αPD1-mIL12mut2 treatment were re-challenged with MC38 tumor cells, these mice quickly rejected the tumor (Fig. 6g). They achieved

long-term survival, confirming that αPD1-mIL12mut2 treatment could induce adequate immunological memory protection.

## Discussion
The potent bioactivity of IL-12 in enhancing effector cell cytotoxicity holds great potential for cancer immunotherapy. However, the life-threatening toxicity associated with systemic IL-12 exposure has greatly limited its clinical application. To address this, we developed a CD8⁺T-cell-targeting low-affinity IL-12, fused with anti-PD-1 antibody. First, we generated two IL-12 mutants with significantly attenuated bioactivity against NK cells. Both mutants demonstrated excellent tolerance and profoundly decreased toxicity when administered systemically in vivo. However, these mutants also exhibited proportionally reduced antitumor activity. To enhance the targeting of the IL-12

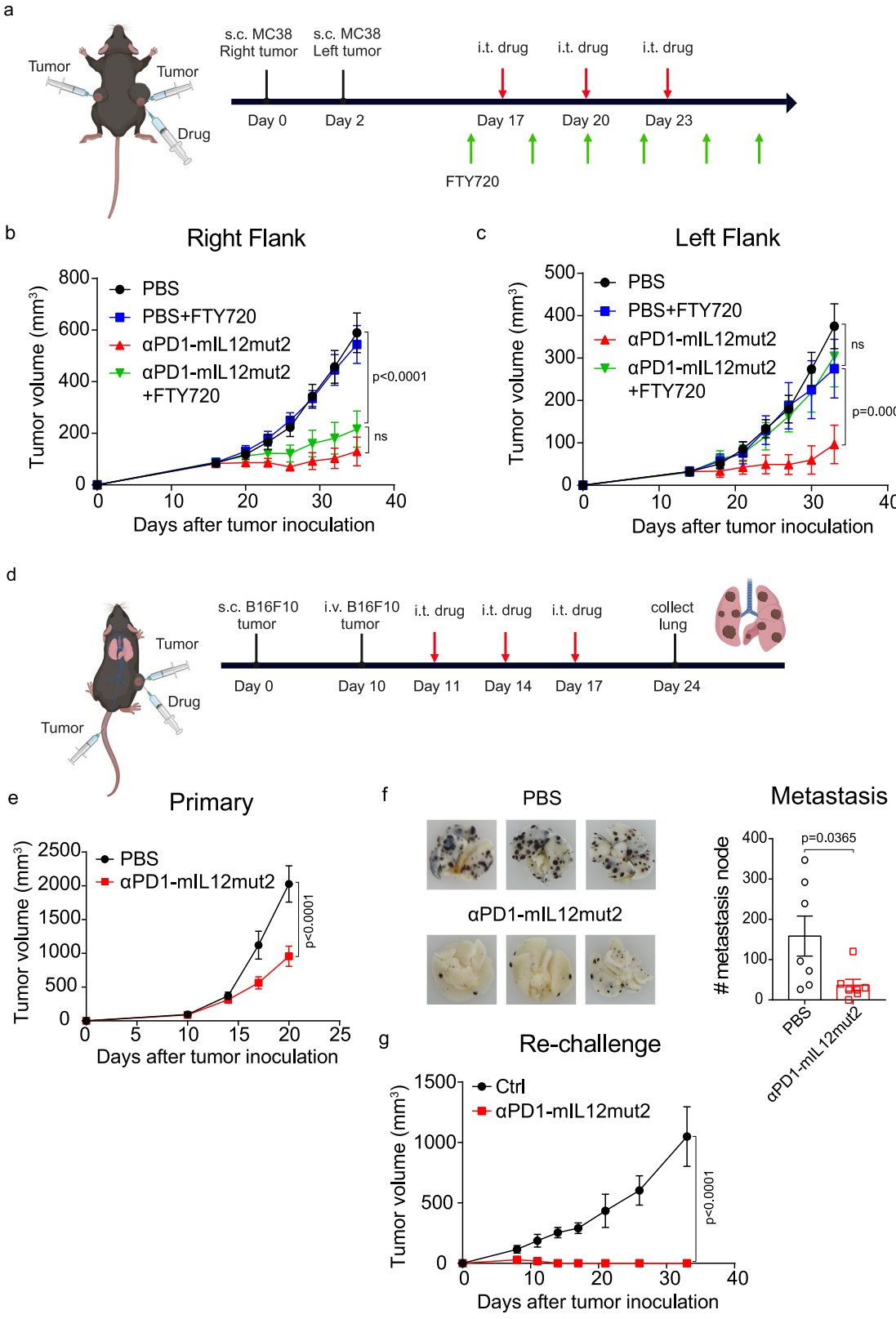

mutant to intratumoral PD-1⁺CD8⁺T cells, we designed a fusion protein called αPD1-mIL12mut2, consisting of a high affinity αPD-1 antibody and a lower affinity IL-12 mutant. We confirmed that αPD1-mIL12mut2 not only effectively retained in tumors but also preferentially bound to intratumoral PD-1⁺CD8⁺T cells. More importantly, αPD1-mIL12mut2 specifically activated intratumoral PD-1⁺CD8⁺T cells with greatly enhanced bioactivity through αPD-1-mediated *cis*-binding. Moreover,

αPD1-mIL12mut2 treatment effectively controlled both primary and distal tumor growth and suppressed metastasis. Altogether, we have developed a next-generation IL-12-based immunotherapeutic drug that elicits robust systemic antitumor immune responses with limited toxicity.

The systemic application of IL-12 is associated with severe toxicity in both mice and humans, yet the mechanism underlying

**Fig. 6 | Anti-PD1-mIL12mut2 treatment induces systemic antitumor immune responses. a** C57BL/6 mice (*n* = 6/group) were inoculated with 5 ×10⁵ MC38 tumor cells at the right flank and 2.5 ×10⁵ MC38 tumor cells at the left flank two days later. The tumor on the right was intratumorally treated with PBS or 10 μg αPD1-mIL12mut2 on days 17, 20, and 23. Mice were intraperitoneally injected with 25 μg FTY720 one day before treatment and then once every other day to block T cells exiting from the lymph node. This picture was produced in BioRender.com. **b** The tumor growth curve on the right flank. **c** The tumor growth curve on the left flank. **d** C57BL/6 mice (*n* = 7/group) were subcutaneously inoculated with 2×10⁵ B16F10 tumor cells and injected with 3×10⁶ B16F10 tumor cells through the tail vein on day 10. The subcutaneous tumor was intratumorally treated with PBS or 50 μg αPD1-mIL12mut2 on days 11, 14, and 17. On day 24, the mice were sacrificed, and the lungs were collected. This picture was produced in BioRender.com. **e** The subcutaneous B16F10 tumor growth curve. **f** The three representative pictures of all seven samples in each group and number of B16F10 tumor metastatic nodes in the lung. **g** The tumor growth curve of the re-challenged mice. C57BL/6 wild-type mice (Ctrl) or the mice with complete MC38 tumor regression (*n* = 4/group) after αPD1-mIL12mut2 treatment were subcutaneously inoculated with 2.5 ×10⁶ MC38 tumor cells 120 days later. All data are shown as mean ± SEM from two to three independent experiments. The *P* value was determined by unpaired t test (**f**) or two-way ANOVA (**b**, **c**, **e**, **g**).

these side effects has remained elusive. It was reported that IFN-γ, a central downstream effector molecule of IL-12, is responsible for IL-12-induced weight loss, leukopenia, and liver damage[11,12,32]. Both NK cells and activated CD8⁺T cells can produce substantial amounts of IFN-γ in response to IL-12 stimulation, potentially contributing to the elevated serum IFN-γ levels following IL-12 administration. We and other researchers have observed a substantial decrease in serum IFN-γ level when NK cells were depleted during IL-12 treatment[17], confirming that NK cells are the primary source of elevated IFN-γ in the serum. Although NK cells appear to be mainly responsible for IL-12-related side effects with weak antitumor activity against primary tumors, it has also been reported that NK cells play a crucial role in suppressing tumor metastasis in IL-12 treatment[33]. Therefore, the precise role of IL-12 in different tumor models remains to be elucidated. In certain models, tissue-resident NK cells have been shown to promote DC-CD8⁺T cell interactions, enhancing T-cell function and thereby supporting a more effective tumor response to IL-12 therapy[34]. Additional tumor models are needed to comprehensively explore the diverse mechanisms underlying IL-12-induced antitumor activities.

Numerous strategies have been adopted to ensure that IL-12 functions locally within tumors with mitigated systemic toxicity. To systemically deliver IL-12 with reduced toxicity, our research group and others previously proposed the concept of "pro-IL-12", which exhibits limited activity until cleaved in tumor tissue[16,17]. Although pro-IL-12 effectively blocks IL-12 activity in its primitive state, its ability to restore IL-12 activity in vivo relies highly on the expression of specific proteases in various tumor types. Recently, an alternative approach emerged, which uses a T-cell biased IL-12 agonist designed to preferentially activate CD8⁺T cells over NK cells[18]. While these approaches primarily aim to alleviate severe toxicity in the periphery, they often lack the necessary tumor-targeting capability to promote effective tumor retention, potentially limiting antitumor efficacy. Moreover, the overall blocking effect of these modified IL-12 molecules was limited, and detectable toxicity remained a concern, particularly when considering a range of therapeutic doses. In addition, some studies have reported targeting IL-12 into tumor tissue using antibodies[19,20]. However, the high affinity of IL-12 often leads to its binding with peripheral NK cells prior to reaching the TME, resulting in excessive consumption, severe toxicity, and disruption of the tumor-targeting efficacy of these immuno-cytokines. Consequently, low-affinity IL-12, designed explicitly for tumor targeting, has emerged as a promising alternative for delivering of IL-12 into tumors.

As mentioned earlier, IL-12 has often been fused with antibodies targeting tumor or stromal cells to enhance tumor retention. In our study, we propose an alternative approach that targets intratumoral CD8⁺T cells via an anti-PD-1 antibody. First, it is worth noting that tumor-infiltrating CD8⁺T cells uniquely expressed significantly higher levels of PD-1 than peripheral T or NK cells. More importantly, the bioactivity of the IL-12 mutant can be selectively and substantially restored, specifically on PD-1⁺CD8⁺T cells, through a *cis*-binding interaction facilitated by anti-PD-1 antibodies. This innovative approach ensures that only intratumoral PD-1⁺CD8⁺T cells are effectively activated by αPD1-mIL12mut2, leading to robust antitumor immune responses.

Throughout tumor progression, naïve T cells are initially primed in tumor-draining lymph nodes (TDLNs) and then migrate into tumors. The importance of T cells in the TDLN for effective antitumor effects is evident, as the complete abrogation of anti-PD-1 antibody treatment efficacy was observed in the presence of FTY720[35]. Furthermore, a distinct subpopulation of tumor-specific memory T cells in the TDLNs has been identified as bona fide responders to PD-1/PDL1 blockade[36]. Concurrently, a reservoir of stem-like TCF-1⁺CD8⁺T cells in TDLN has been shown to migrate into tumors progressively, preserving antitumor immune responses[37,38]. However, whether preexisting TILs can be sufficiently reactivated within the TME to achieve robust antitumor activity remains unclear. In this study, we observed that αPD1-mIL12mut2 remained active against established tumors even when T-cell trafficking into tumors was blocked by FTY720 during treatment. Together, these findings suggest that this fusion protein can target preexisting TILs to enhance antitumor immunity.

In the early stage of tumor progression, when tumors are not well-established, primed T cells migrate from the TDLN into tumors. During the early phase, the limited number of CD8⁺T cells in tumors may be insufficient to mediate effective antitumor responses[39,40]. Conversely, a substantial number of T cells have infiltrated into established tumors. Thus, the migration of T cells from the TDLN into tumors might be unnecessary for successful immunotherapy. In conclusion, the fusion protein αPD1-mIL12mut2 offers more than simple PD-1/PDL1 blockade antibodies. It can potently reactivate TILs beyond just alleviating immunosuppression, resulting in robust TILs-mediated antitumor immune responses.

Although this T-cell-targeting IL-12 fusion protein can preferentially activate tumor-infiltrating CD8⁺T cells and induce robust antitumor immune responses with limited side effects, it may have several potential limitations. First, as tumor-infiltrating CD8⁺T cells were identified to be primarily responsible for the antitumor effect of αPD1-mIL12mut2 (Fig. 5b), tumors with abundant T-cell infiltration, known as "hot" tumors, might respond to αPD1-mIL12mut2 treatment. However, "cold" or immune-excluded tumors lacking enough T-cell infiltration may be resistant to αPD1-mIL12mut2 treatment. Moreover, PD-1 expression on CD8⁺T cells is also essential for the treatment efficacy of αPD1-mIL12mut2 as both the PD-1 targeting effect and αPD-1-mediated *cis*-binding lead to the restoration of IL-12 stimulatory signaling to CD8⁺T cells. Therefore, additional immunotherapies[41] to promote CD8⁺T cell activation[10] and infiltration[42] in combination with αPD1-mIL12mut2 treatment holds the key to overcoming tumor resistance and successfully treating more types of cancer.

Another issue is the incomplete investigations of the immune-related adverse effects (irAEs) of αPD-1 therapy. PD-1/PD-L1 blockades alleviate immunosuppression and reactivate the immune system, which can lead to the development of irAEs that may affect and even cause damage to multiple tissues and organs[43]. The irAEs of αPD-1 therapy commonly occur in clinical cancer patients but do not often occur in mouse tumor models. Although this study observed no severe irAEs after αPD1-mIL12mut2 treatment within this study, awareness

should be raised, and more comprehensive investigations are urgently needed, especially for some tumor-bearing hosts that might have higher PD-1 expression outside the TME.

In conclusion, low-affinity IL-12 not only circumvents unnecessary peripheral consumption to mitigate severe toxicity but also complements the predominant role of the αPD-1 antibodies, optimizing their tumor-targeting effect. The αPD-1 antibodies effectively bind to intratumoral PD-1⁺CD8⁺T cells and directly anchor IL-12 on these effector cells. Our αPD1-mIL12mut2 fusion protein can induce robust systemic antitumor responses with precision and minimal side effects, showing great potential for clinical translation.

# Methods

## Mice
Female C57BL/6 mice about 6-8 weeks old were purchased from SPF Biotechnology. $Rag1^{-/-}$ mice were purchased from the Model Animal Research Center, Nanjing University. $Pdcd1^{-/-}$ mice were purchased from the Animal Management Center, Institute of Biophysics. NOD/SCID/IL2rγ$^{null}$ (NSG) mice were maintained internally. All mice were maintained under specific pathogen-free conditions with 10/14 dark/light cycle, 20–26 °C, and 30–70% humidity. Animal care and experiments were carried out under institutional protocol and guidelines. All studies were approved by the Animal Care and Use Committee of the Institute of Biophysics.

## Cell lines and reagents
MC38, B16F10, and A549 cell lines were purchased from the American Type Culture Collection. The MC38-EGFR5 cell line was obtained from transduction by lentivirus encoding the mutant mouse EGFR gene. All the cells were cultured in 5% CO₂ at 37 °C and maintained in Dulbecco's modified Eagle medium (DMEM) supplemented with 10% heat-inactivated FBS, 2mM L-glutamine, 100 U/ml penicillin-streptomycin. Freestyle 293 F cell (R79007) was purchased from Invitrogen and cultured in SMM 293-TII medium. HEK-Blue™ IL-12 cell line was purchased from InvivoGen and cultured in Dulbecco's modified Eagle medium (DMEM) supplemented with 10% heat-inactivated FBS, 2 mM L-glutamine, 50 U/ml penicillin, 50 μg/ml streptomycin, 100 μg/ml Normocin™ (ant-nr-1).

FTY720 was purchased from Sigma-Aldrich. Anti-CD8 antibody (TIB210), anti-CD4 antibody (GK1.5), anti-NK1.1 antibody (PK136), anti-IFN-γ (XMG1.2) and FcγR II/III blocking antibody (2.4G2) were produced in-house. Gene sequences encoding anti-Mouse PD-1 antibody (J43) and anti-Human PD-1 antibody (Keytruda) were used for fusion protein construction and antibody production. All gene sequences of fusion proteins were cloned into a pEE12.4 vector and transfected into 293F cells. The supernatant was purified by Protein-A affinity chromatography (GE Healthcare) according to the established protocol.

The abbreviations used to designate the fusion proteins used in this study can be found in Table 1.

**Table 1 | Abbreviations for fusion proteins**

| Fusion protein forms | Abbreviation |
| --- | --- |
| Mouse IL-12 wild type | mIL12wt |
| Anti-PD-1-mouse IL-12 wild type | αPD1-mIL12wt |
| Anti-PD-1-mouse IL-12 mutant-1 | αPD1-mIL12mut1 |
| Anti-PD-1-mouse IL-12 mutant-2 | αPD1-mIL12mut2 |
| Human IL-12 wild type | hIL12wt |
| Anti-PD-1-human IL-12 wild type | αPD1-hIL12wt |
| Anti-PD-1-human IL-12 mutant-1 | αPD1-hIL12mut1 |
| Anti-PD-1-human IL-12 mutant-2 | αPD1-hIL12mut2 |
| Anti-human PD-1-human IL-12 mutant-2 | αhPD1-hIL12mut2 |

## Tumor growth and treatment
Mice were subcutaneously inoculated on the right flank with $5*10^5$ MC38 or $2*10^5$ B16F10 tumor cells. For the double-flank tumor model, $2.5*10^5$ MC38 tumor cells were subcutaneously inoculated on the left flank of mice. For the experimental pulmonary metastasis model, $3*10^6$ B16F10 tumor cells were intravenously injected into mice through the tail vein one day before fusion protein treatment. After tumors were established, the tumor-bearing mice were randomly grouped. Mice were intraperitoneally or intratumorally treated with mIL12wt, αEGFR-mIL12mut2, αPD1-mIL12mut2, and αPD1-mIL12wt, respectively. For cell depletion, mice were intraperitoneally injected with 400 μg αNK1.1 antibody, 200 μg αCD4 antibody, or 200 μg αCD8 antibody, respectively, one day before the first dose of fusion protein treatment and then once every four days for three times. For IFN-γ neutralization, mice were intraperitoneally injected with 500 μg αIFN-γ antibody one day before the first treatment of fusion proteins and then once every four days for three times. For FTY720 blockade, mice were intraperitoneally injected 25 μg FTY720 one day before the first treatment of fusion proteins and then once every other day for seven times. The tumor volume was measured every two days and calculated as length*width*height/2. The humane endpoints are defined as no greater than 1000 mm³ for the MC38 tumor model and no greater than 2000 mm³ for the B16F10 tumor model.

## Toxicity evaluation
Tumor-bearing mice were treated with different fusion proteins. The mouse weight at the beginning of the treatment was set as the initial body weight, and the mouse weight measured after that was set as the measured body weight. The weight change was calculated as (measured body weight − initial body weight)/initial body weight*100%. The blood sample was collected at different time points after treatment, and the serum was separated. The concentration of cytokines and transaminases in serum was measured by Cytometric Bean Array and Transaminase Assay Kit respectively.

## HEK-Blue™ IL-12 reporter assay
HEK-Blue™ IL-12 cell line was used to detect the activity of mouse or human IL-12 in vitro. $5*10^4$ HEK-Blue™ IL-12 cells were incubated with different concentrations of proteins for 24 h in 5% CO₂ at 37 °C. 20 μl supernatant was added into 180 μl QUANTI-Blue™ (rep-qb1) and incubated for 3 h. The SEAP activity was assessed by reading the OD at 620-655 nm.

HEK-Blue™ IL-12 cell line was transduced with lentivirus encoding the mouse $pdcd1$ gene, and the HEK-mPD-1 cell line was obtained after puromycin selection. The mouse PD-1 expression was detected by Flow Cytometry. For PD-1 blockade, different concentrations of anti-PD-1 antibodies were added into the incubation system of HEK-mPD-1 cells and fusion proteins.

## Measurement of p-STAT4 and IFN-γ induction
The spleen from wild-type or $pdcd1^{-/-}$ C57BL/6 mice were isolated. Red blood cells were lysed using ACK lysis buffer, and a single cell suspension was generated. Splenocytes were activated with 2.5 μg/ml αCD3, 2.5 μg/ml αCD28, and 100 IU/ml recombinant IL-2 for 48 h at 37 °C. CD8⁺T cells were sorted using a mouse CD8⁺T cell isolation kit (BioLegend). NK cells were sorted from fresh splenocytes using a mouse NK cell isolation kit ((BioLegend)). For the p-STAT4 signal, CD8⁺T cells or NK cells were incubated with different proteins for 30 minutes at 37 °C. Cells were stained with anti-p-STAT4 Y693 APC (4LURPIE, Invitrogen) and analyzed by Flow Cytometry. For IFN-γ induction, CD8⁺T cells or NK cells were incubated with different proteins for 48 h at 37 °C. The supernatant was assessed for IFN-γ using CBA kit (BD).

### *Cis*-binding assay

Pre-activated wild-type or *pdcd1*[−/−] CD8[+]T cells were sorted using CD8[+]T cell isolation kit (BioLegend). The wild-type CD8[+]T cells were labeled with Cell trace violet (CTV). An equal number of CTV-labeled wild-type CD8[+]T cells and unlabeled *pdcd1*[−/−] CD8[+]T cells were mixed and incubated with αEGFR-mIL12mut2 or αPD1-mIL12mut2 for 30 min at 37 °C. Cells were stained with anti-p-STAT4 and analyzed by Flow Cytometry.

### Experimental pulmonary metastases model

C57BL/6 mice were subcutaneously injected with $2*10^5$ B16F10 tumor cells. On day 10, $3*10^6$ B16F10 tumor cells were intravenously injected into mice through the tail vein. On day 11, the subcutaneous tumor was intratumorally treated with 50 μg αPD1-mIL12mut2 three times. On day 24, the mice were sacrificed, and the lungs were removed and fixed. The metastatic nodules in the lungs were counted.

### Flow cytometry

Tumor tissues were collected, cut into small pieces, and re-suspended in digestion buffer (RPMI-1640 medium with 1 mg/mL type IV collagenase and 100 μg/mL DNase I). Tumors were digested for 45 minutes at 37 °C and then passed through 70 μm cell strainer to make single-cell suspensions. Single-cell suspensions were incubated with FcγR II/III blocking antibody (2.4G2) and stained with specific antibodies followed by established protocol. For intracellular IFN-γ and Foxp3 staining, samples were fixed, permeabilized, and stained with anti-mouse IFN-γ or anti-mouse Foxp3 antibodies. LIVE/DEAD fixable yellow dye (Thermo Fisher Scientific) was used to exclude dead cells. Samples were analyzed on a Fortessa flow cytometer (BD), and data were analyzed by FlowJo software (Treestar).

### Quantitative flow cytometry

For establishing a standard curve, we followed the methodology outlined in the Quantibrite™ PE Phycoerythrin Fluorescence Quantitation Kit protocol (BD, 340495), which employs beads conjugated with four standard levels of PE fluorescence. 10,000 PE beads were detected by flow cytometry at a specific voltage setting, which was subsequently applied for the quantification of markers. The distinct and well-separated four-peak signals of PE beads correspond to beads carrying varying numbers of PE, categorized as Low, Med Low, Med High, and High levels. The standard curve was derived through linear regression of Log10 PE/Bead and Log10 Geo Mean, based on the equation: $Y = 0.9993*X − 0.2081$, where $Y$ represents Log10 Geo Mean and $X$ represents Log10 PE/Bead. Receptor quantification per cell was achieved using PE-conjugated antibodies specific to the receptors of interest. Furthermore, a series of antibody concentration titrations were conducted to confirm saturated binding to the target cell membrane receptors. The mean count of receptors on cells was quantified at the saturation concentration according to the standard curve.

### Statistical analysis

Data are shown as the mean ± SD or mean ± SEM. All analyses were performed using GraphPad Prism statistical software (GraphPad Software Inc., San Diego, CA). The *P* value is determined by one-way, two-way ANOVA with multiple comparison, or two-tailed unpaired *t* test. Differences were considered statistically significant if $P < 0.05$.

### Reporting summary

Further information on research design is available in the Nature Portfolio Reporting Summary linked to this article.

## Data availability

Source data are available online for Figs. 1–6 and Supplementary Figs. 1–6. All other data that support the findings of this study are available from the corresponding author upon request. Source data are provided with this paper.

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

## Acknowledgements

We thank all lab members and the support from the animal facility of the Institute of Biophysics, Chinese Academy of Sciences. This work was supported by funding from the Chinese Academy of Sciences (KFJ-STS-ZDTP-062) and (XDA12020212) to H.P. and National Key S&T Special project of China (2018ZX1030140402) to H.P.

## Author contributions

Conceptualization: Zhuangzhi Zou, Hua Peng, Yang-Xin Fu; Methodology: Zhuangzhi Zou, Jiao Shen, Diyuan Xue; Investigation: Zhuangzhi Zou, Jiao Shen, Diyuan Xue, Hongjia Li, Longxin Xu, Weian Cao, Wenyan Wang; Formal Analysis: Zhuangzhi Zou, Jiao Shen, Diyuan Xue, Hongjia Li, Longxin Xu, Hua Peng, Yang-Xin Fu; Writing-Original Draft: Zhuangzhi Zou; Writing-Review & Editing: Zhuangzhi Zou, Jiao Shen, Hua Peng, Yang-Xin Fu; Supervision: Hua Peng, Yang-Xin Fu.

## Competing interests

The authors declare no competing interests.
