## [Peer Review File · Nature Communications]

Anti-PD-1 *cis*-delivery of low-affinity IL-12 activates intratumoral CD8⁺T cells for systemic antitumor responsesREVIEWER COMMENTS

Reviewer #1 (expert in translational immunology and cancer therapy):

Anti-PD-1 cis-delivery of low-affinity IL-12 activates intratumoral CD8⁺T cells for systemic antitumor responses

Zhu et al has presented a well-written, clear manuscript detailing an approach to deliver low-affinity IL-12 to tumor infiltrating PD1⁺ CD8 T cells. Although approaches for using IL-12 to activate intratumoral CD8T cells have been previously described, the method and clarity of data are novel. This reviewer's impression of the data is that a sub-optimal IL-12, which is still capable of binding to IL12R, can be delivered at a high enough local concentration to activate T cells safely. The new finding is that targeting the lymphocyte, and not the tumor, with the IL-12 is important for activity.

Minor concerns:

Line 149-151, "concentration was higher in tumor than other tissues" not supported, it was ns different from Liver and spleen. 151-152 "preferentially accumulate in tumors" is not supported by the data.

Labeling murine Mutant IL-12 (mIL-12) can be confused with mouse IL-12 especially in the context of calling human IL-12 hIL-12. Although it is explained in the text, this reviewer was confused for a moment that mIL-12 meant mouse IL-12, not mutant-2 IL-12. Consider a different labeling strategy to keep it clear, maybe mm (mouse mutant) or mIL-12 Mut-1 and hIL-12 Mut-2.

Please report concentrations as nM, some use ng/mL and other graphs use nM. Helpful For comparing concentrations across multiple experiments

Figure 6 could use some schemas to clarify experimental design.

Line 267: Clarify the PD1⁻ subset, is that PD1⁻ Tim-3⁺ +/- or is it PD1⁻ Tim-3⁻?

Figure s4 If PD1⁺/Tim-3⁺ cells represent exhausted cells, why isn't there reduced IFN γ in PBS treated mice compared to PD1⁻Tim3⁻ cells? Line 275: Do not use the term, re-invigorate, as the cells all look similar in their functionality as far as IFN-g directly ex vivo is concerned.

Figure 6A/B/E, show the individual growth curves and/or indicate which or how many mice completely rejected the tumor, presumably control mice are tumor naïve?

Reviewer #2 (expert in protein engineering and translational immunology):

Summary:

This manuscript by Zou and colleagues describes the therapeutic effects of a fusion protein that comprises an engineered affinity-impaired IL-12 cytokine and an anti-PD-1 antibody. The investigators aim to combine targeted tumor-infiltrating CD8⁺ T cell activation with immune checkpoint blockade as an antitumor strategy. The treatment was found to achieve efficacy through cis delivery of IL-12 to exhausted Tregs and was shown to exhibit antitumor effects at the primary site of disease as well as in distal sites and metastases. Moreover, the authors demonstrated attenuated toxicity compared to treatment with the native IL-12 cytokine. Overall, the paper is timely and interesting, providing both mechanistic insights and therapeutically promising findings highly relevant to the field of cancer immunotherapy. However, there are several technical limitations that need to be addressed, and in particular, the authors need to demonstrate the advantage in therapeutic window compared to WT IL-12 fusion proteins. Also, the writing needs to be improved for clarity. Specific points of revision are noted below. Revision would be required for this work to be appropriate for Nature Communications.

Specific Points:

- 1) The manuscript requires significant revision with respect to phrasing and grammar.
- 2) On-cell titrations shown in Fig. 2C look very weak and are not saturating. Moreover, the baselines are different and Mut-1 looks to be higher affinity than WT. Do the authors have an explanation for this? Are trafficking effects impacting the results?
- 3) Anti-PD-1-mIL-12 is described as a homodimer, but this seems to be a heterodimeric Fc. The authors should address this in the text.

- 4) Figures 2E and 2F should have synchronized formatting.
- 5) Figures 1 and 2 should be in color.
- 6) The binding affinity for the anti-PD-1 antibody should be provided and compared to that of the attenuated IL-12 for the receptor signaling complex.
- 7) Figure 3A should be presented as a titration.
- 8) For the engineered mPD-1-expressing HEK cells, the number of receptors per cell for PD-1, IL-12Rb1, and IL-12Rb2 need to be quantified. How do these numbers reflect the corresponding number of receptors on tumor cells?
- 9) Figure 3F should include a WT IL-12 fusion protein as a control.
- 10) Figures 4B-D should include comparisons with WT IL-12.
- 11) In the MC38-EGFR5 model, what is the relative expression of mouse EGFR versus mPD-1 on tumor cells?
- 12) How effective is WT IL-12 at 5 ug doses? Comparison between the efficacy of aPD-1-mIL-12 and aPD-1-IL-12 in tumor models side by side with toxicity data is needed.

Reviewer #3 (expert in biomedical engineering and IL-12-based therapies):

This manuscript describes the development of a novel immunocytokine comprised of anti-PD-1 fused to a low-affinity IL-12 construct. The goal of the study was to target IL-12, whose clinical development has been stymied by serious toxicities, to TILs via anti-PD-1 while minimizing signaling in peripheral lymphocytes, particularly NK cells, via the mutated IL-12. Impressively, systemic administration of the immunocytokine did not result in toxicities that are commonly associated with IL-12. Furthermore, studies demonstrate that anti-PD-1 engagement resulted in cis-binding and subsequent signaling of low affinity IL-12. Although the authors claim an abscopal effect, this is debatable since an intratumoral injection results in systemic uptake. This strategy requires TILs expressing PD-1 and would not appear to be effective in cold, excluded tumors. This caveat should be added. Another limitation of the manuscript is that it does not address immune related toxicities of anti-PD-1 which are not often reproduced in mice. The fusion of IL-12 is likely to exacerbate autoimmune related toxicities in humans. In addition, the timing of the body weight loss and toxicity is curious. Systemic IL-12 treatments typically result in a spike in IFNg within 12-36 hrs which is associated with weight loss and elevated liver enzymes. It is not clear why toxicity is delayed until day 7-9 after treatment.

A number of additional minor concerns should be addressed:

- Line 60 – the statement that ‘numerous strategies have been proposed to reduce side effects by local delivery’ should include a reference
- Line 72 – ‘PD-1 has much higher expression inside TME...’ should include a reference
- Line 80 – T cells are not ‘naïve’; only CD4 and CD8 markers were used
- Line 92 – the disposability of NK cells is model dependent; authors should not claim that NK cells are not important in IL-12 immunotherapy
- Line 136 – no detail about anti-PD-1, either the mouse or human version
- Line 137 – mIL-12 is the standard nomenclature for murine IL-12. It is strongly suggested that the authors do not use mIL-12 to mean mutated IL-12. This is very confusing.
- Line 150 – the statement that the concentration of aPD-1-mIL-12 was obviously higher in the tumor is not correct. Fig S2D shows similar concentrations in the liver and spleen.
- Line 152 – data are not convincing that aPD-1-mIL-12 ‘preferentially accumulates in tumors
- Line 175 – why doesn’t aPD-1-mIL-12 result in higher signaling than aEGFR-mIL-12 in HEK-mPD-1 cells (Fig S3C)?
- Line 202 – mice also treated on days 17 and 20 (not just day 14)
- Line 258 – Fig S4F does not appear to demonstrate that FTY720 diminishes T cells in the periphery
- Line 281 – intratumoral injections, like s.c. injections result in systemic distribution
- Line 572 – for all graphs reporting ‘Days after treatment’; please clarify is this after 1, 2,3 etc injections?
- Line 601 – the ‘Ctrl’ group in Fig 3C is not defined
- Line 713 – when after treatment were tumors harvested?

RESPONSE TO REVIEWERS' COMMENTS

Reviewer #1 (expert in translational immunology and cancer therapy):

Anti-PD-1 cis-delivery of low-affinity IL-12 activates intratumoral CD8⁺T cells for systemic antitumor responses. Zhu et al has presented a well-written, clear manuscript detailing an approach to deliver low-affinity IL-12 to tumor infiltrating PD1⁺ CD8 T cells. Although approaches for using IL-12 to activate intratumoral CD8T cells have been previously described, the method and clarity of data are novel. This reviewer's impression of the data is that a sub-optimal IL-12, which is still capable of binding to IL12R, can be delivered at a high enough local concentration to activate T cells safely. The new finding is that targeting the lymphocyte, and not the tumor, with the IL-12 is important for activity.

Minor concerns:

Line 149-151, "concentration was higher in tumor than other tissues" not supported, it was ns different from Liver and spleen. 151-152 "preferentially accumulate in tumors" is not supported by the data.

Thanks to the reviewer's comment. We repeated this experiment to track the tissue distribution of α PD-1-IL-12m in different conditions. We treated mice with 20 μ g α PD-1-IL-12m, the effective dose in this tumor model, and collected mouse organs and tissues 24 hours later. As shown in Fig R1, the concentration of α PD-1-IL-12m in the tumor is significantly higher than in other tissues, including the liver and spleen. This result confirms that α PD-1-IL-12m is preferentially accumulated in tumors. We have added this result in the revised manuscript.

Fig R1. MC38 tumor-bearing mice (n = 5) were i.p. injected with 20 μ g α PD-1-IL-12m and sacrificed 24h after treatment. Mice tissues were collected and homogenized. The concentration of fusion protein in the supernatant was measured and normalized by ELISA. The P value was determined by one-way. *P < 0.05, **P < 0.01, ***P < 0.001, ****P < 0.0001.

Labeling murine Mutant IL-12 (mIL-12) can be confused with mouse IL-12 especially in the context of calling human IL-12 hIL-12. Although it is explained in the text, this reviewer was confused for a moment that mIL-12 meant mouse IL-12, not mutant-2 IL-12. Consider a different labeling strategy to keep it clear, maybe mm (mouse mutant) or mIL-12 Mut-1 and hIL-12 Mut-2.

Thanks to the reviewer's suggestion. In the following table, we used new nomenclature to clarify the different forms of fusion proteins. Because mouse IL-12 Mut-2 was used in most experiments, we simplified the wild-type mouse IL-12 as IL-12 and mouse IL-12 Mut-2 as IL-12m.

Table 1. Abbreviation for fusion proteins

Fusion protein forms	Abbreviation
mouse IL-12 wild type	IL-12
anti-PD-1-mouse IL-12 wild type	α PD-1-IL-12wt
anti-PD-1-mouse IL-12 Mut-1	α PD-1-IL-12m1
anti-PD-1-mouse IL-12 Mut-2	α PD-1-IL-12m
human IL-12 wild type	hIL-12
anti-PD-1-human IL-12 wild type	α PD-1-hIL-12wt
anti-PD-1-human IL-12 Mut-1	α PD-1-hIL-12m1
anti-PD-1-human IL-12 Mut-2	α PD-1-hIL-12m
anti-human PD-1-human IL-12 Mut-2	α hPD-1-hIL-12m

Please report concentrations as nM, some use ng/mL and other graphs use nM. Helpful For comparing concentrations across multiple experiments

Thanks to the reviewer's suggestion. The concentration in this manuscript is unified as nM.

Figure 6 could use some schemas to clarify experimental design.

Thanks to the reviewer's suggestion. We have added the schemas in Fig.6 in the revised manuscript.

Line 267: Clarify the PD1- subset, is that PD1- Tim-3 +/- or is it PD1- Tim-3-?

Thanks to the reviewer's comment. As shown in Fig R2, all PD-1⁺CD8⁺T cells in this manuscript were TIM-3 negative. The PD-1⁺CD8⁺T cells in tumor tissues barely expressed TIM-3, no matter whether MC38 tumor-bearing mice were treated with PBS or α PD-1-IL-12m. This result is consistent with a previous report¹.

Fig R2. MC38 tumor-bearing mice were i.p. injected with PBS or 20 μ g α PD-1-IL-12m. Tumors were collected 48 hours after the second treatment, and the expression of PD-1 and TIM-3 on CD8⁺T cells was detected. FMO: Fluorescence Minus One.

Figure s4 If PD1⁺/Tim-3⁺ cells represent exhausted cells, why isn't there reduced IFN γ in PBS treated mice compared to PD1⁻Tim3⁻ cells?

Thanks to the reviewer's comment. In Fig s4I, the IFN- γ -producing CD8⁺T cells were detected by *in vivo* intracellular cytokines staining but not an *ex vivo* assay. We have made this figure legend more evident. Brefeldin A (BFA) (420601, Biolegend) was intraperitoneally injected into mice six hours before collecting tumor tissues to prevent cytokines secretion². Although PD-1⁺TIM-3⁺CD8⁺T cells are well recognized as exhausted cells with impaired effector function, especially IFN- γ production³, we found that all subpopulations of CD8⁺T cells (including PD-1⁺TIM-3⁺CD8⁺T) in PBS treated mice produced low level of IFN- γ (Fig s4I). A previously published article also proved that the ability of IFN- γ production of CD8⁺T cells was shallow with no difference among PD-1⁻TIM-3⁻, PD-1⁺TIM-3⁻ and PD-1⁺TIM-3⁺ subpopulations *in vivo*⁴, indicating that tumor-infiltrating CD8⁺T cells are not active in producing IFN- γ . Under this condition, it is not easy to compare the function of these subpopulations. When these subpopulations of CD8⁺T cells were sorted and re-stimulated *ex vivo* by PMA and ionomycin¹ or cognate antigen⁵, the functionally impaired PD-1⁺TIM-3⁺CD8⁺T cells produced largely reduced cytokines, including IFN- γ , compared to other subpopulations. As PD-1 and TIM-3 are commonly accepted to define exhausted T cells bearing a stable and irreversible epigenetic landscape⁶, these two markers were used to define dysfunctional T cells in this manuscript.

Line 275: Do not use the term, re-invigorate, as the cells all look similar in their functionality as far as IFN-g directly *ex vivo* is concerned.

Thanks to the reviewer's suggestion. PD-1⁺TIM-3⁺CD8⁺T cells are well recognized as terminal exhausted T cells with a stable and irreversible epigenetic landscape⁶. Therefore, the enhanced IFN- γ production of PD-1⁺TIM-3⁺CD8⁺T cells by α PD-1-IL-12m treatment should be defined as "partially restored" but not "reinvigorated" effector function.

Figure 6A/B/E, show the individual growth curves and/or indicate which or how many mice completely rejected the tumor, presumably control mice are tumor naïve?

Thanks to the reviewer's comment. The individual tumor growth curve of mice is shown in Fig R3. The α PD-1-IL-12m treatment significantly delayed MC38 tumor growth. Nevertheless, mice in all groups died finally.

In Fig 6E (6G revised), the control mice are WT naïve mice. The tumor-free mice were chosen from the previous MC38-tumor-bearing mice whose tumors had been completely eradicated for over two months by α PD-1-IL-12m treatment.

Fig R3. The individual growth curves of both the right (A) and the left flank (B) MC38 tumor. Related to Fig. 6A (6B revised) and 6B (6C revised).

Reviewer #2 (expert in protein engineering and translational immunology):

Summary:

This manuscript by Zou and colleagues describes the therapeutic effects of a fusion protein that comprises an engineered affinity-impaired IL-12 cytokine and an anti-PD-1 antibody. The investigators aim to combine targeted tumor-infiltrating CD8+ T cell activation with immune checkpoint blockade as an antitumor strategy. The treatment was found to achieve efficacy through cis delivery of IL-12 to exhausted Tregs and was shown to exhibit antitumor effects at the primary site of disease as well as in distal sites and metastases. Moreover, the authors demonstrated attenuated toxicity compared to treatment with the native IL-12 cytokine. Overall, the paper is timely and interesting, providing both mechanistic insights and therapeutically promising findings highly relevant to the field of cancer immunotherapy. However, there are several technical limitations that need to be addressed, and in particular, the authors need to demonstrate the advantage in therapeutic window compared to WT IL-12 fusion proteins. Also, the writing needs to be improved for clarity. Specific points of revision are noted below. Revision would be required for this work to be appropriate for Nature Communications.

Specific Points:

1) The manuscript requires significant revision with respect to phrasing and grammar. Thanks to the reviewer's suggestion. The manuscript has been further revised and polished carefully.

2) On-cell titrations shown in Fig. 2C look very weak and are not saturating. Moreover, the baselines are different and Mut-1 looks to be higher affinity than WT. Do the authors have an explanation for this? Are trafficking effects impacting the results?

Thanks to the reviewer's comment. When the concentration of these fusion proteins continued to increase, the MFI in all groups increased sharply and still could not reach saturation point even in an extremely high concentration (~160ug/ml), which is far more than the actual dose we used in mouse models. Moreover, the higher concentration of fusion proteins is more likely to cause non-specific binding. When the fusion proteins were further diluted, the MFI in all groups was very low with the same baseline.

We performed the same binding assay with HEK cells. As shown in Fig R4, WT and Mut-1 had a similar binding affinity, while only Mut-2 showed significantly attenuated binding affinity, consistent with results in Fig 2C. The seemingly higher affinity of Mut-1 than WT in Fig 2C was not evident and not observed in HEK cells. Therefore, the difference between WT and Mut-1 is slight and neglectable.

Besides, unfortunately, the reports about the traffic effects in IL-12 binding with receptors are entirely blank. The incubation of fusion proteins with cells lasted for a short time at 4°C, so the traffic effects are likely to have minimal effect on the results. Nevertheless, we cannot fully exclude the role of traffic effects in this experiment.

Fig R4. HEK cells were incubated with serially diluted WT IL-12, Mut-1, or Mut-2. Protein binding to HEK cells was detected by flow cytometric analysis. MFI: mean fluorescence intensity.

3) Anti-PD-1-mIL-12 is described as a homodimer, but this seems to be a heterodimeric Fc. The authors should address this in the text.

Thanks to the reviewer's suggestion. The description of this fusion protein has been revised as "We constructed a fusion protein by linking one molecule of mouse IL-12 Mut-2 at the C terminal of the Fc domain of anti-PD-1 antibodies, named α PD-1-IL-12m".

4) Figures 2E and 2F should have synchronized formatting.

The Figures 2E and 2F (S2E revised) have been revised to display in synchronized formatting.

5) Figures 1 and 2 should be in color.

Figures 1 and 2 are in color in the revised manuscript.

6) The binding affinity for the anti-PD-1 antibody should be provided and compared to that of the attenuated IL-12 for the receptor signaling complex.

Thanks to the reviewer's suggestion. As shown in Fig R5, the binding affinity of anti-PD-1 (α PD-1, black line) to HEK-mPD-1 reporter cells is much higher than that of the mutated IL-12 (α EGFR-IL-12m, blue line). Adding an anti-PD-1 antibody significantly increases the binding affinity of the mutated IL-12-contained fusion protein (α PD-1-IL-12m, red line). The difference in binding affinity between α PD-1 and α PD-1-IL-12m may result from the interference of IL-12 in the binding of α PD-1 in the form of fusion protein. This result indicates that it is the anti-PD-1 antibody that mainly mediates the enhanced binding of mutated IL-12 fusion protein to the PD-1 positive cells, resulting in restored bioactivity (Fig 3B). The complete loss of this enhanced bioactivity by α PD-1 antibody pre-blockade further emphasized the essential role of PD-1 binding in fixing the bioactivity of mutated low-affinity IL-12 (Fig 3B).

Fig R5. HEK-mPD-1 cells were incubated with serially diluted hIgG1, α EGFR-IL-12m, α PD-1-IL-12m, or anti-PD-1 antibody *in vitro*. Protein binding to HEK-mPD-1 cells was detected by flow cytometric analysis.

7) Figure 3A should be presented as a titration.

We have added the titration data (Fig R5) in Figure 3A of the main manuscript.

8) For the engineered mPD-1-expressing HEK cells, the number of receptors per cell for PD-1, IL-12Rb1, and IL-12Rb2 need to be quantified. How do these numbers reflect the corresponding number of receptors on tumor cells?

Thanks to the reviewer's comments. Tumor cells do not express PD-1 or IL-12Rb1/2 at all. We have measured PD-1, IL-12Rb1, and IL-12-Rb2 expression levels on HEK reporter cells or tumor-infiltrating CD8⁺T cells by flow cytometry. PD-1 has been confirmed with a high expression level on intratumoral CD8⁺T cells (Fig 2B, S2B). As shown in Fig R6, the PD-1 was also highly expressed on HEK-mPD-1 cells, while IL-12Rb1 or IL-12-Rb2 expression was quite low when detected by flow cytometry on both cells. It was reported that Quantitative Flow Cytometry was used to count the absolute number of receptors on the plasma membrane⁷. Because of the extremely low level of IL-12 receptor expression, this method failed to count the number of receptors quantitatively in this case. Nevertheless, we found that α PD-1-IL-12m displayed significantly stronger binding ability on HEK-mPD-1 cells (Fig 3A) and intratumoral CD8⁺T cells than α EGFR-IL-12m (Fig 2E), suggesting that α PD-1 antibodies rather than mutant IL-12 dominated the binding of fusion proteins with effector cell. In lack of the absolute number of PD-1, IL-12Rb1, or IL-12-Rb2, we observed the enhanced binding ability and the restored bioactivity of α PD-1-IL-12m on both HEK-mPD-1 cells (Fig 3A, 3B) and intratumoral CD8⁺T cells (Fig 2E, Fig 5C), and these consistent results together confirmed the α PD-1 antibodies mediated *cis*-binding effect in this study.

Fig R6. The expression of PD-1, IL-12Rb1, and IL-12Rb2 on HEK-mPD-1 cells (A) and intratumoral CD8⁺T cells (B). Black peak: FMO.

9) Figure 3F should include a WT IL-12 fusion protein as a control.

Thanks to the reviewer's suggestion. We have included a WT hIL-12 fusion protein as a control in Figure 3F of the revised manuscript. As shown in Fig R7, α EGFR-hIL-12m2 has similar and low activity on both HEK and HEK-mPD-1 cells, while α PD-1-hIL-12m shows much stronger activity only on HEK-mPD-1 cells but not HEK cells, which is the same to that of hIL-12 on HEK cells.

Fig R7. The activity of wild type hIL-12, α EGFR-hIL-12m, and α PD-1-hIL-12m were measured by HEK or HEK-mPD-1 cells.

10) Figures 4B-D should include comparisons with WT IL-12.

Thanks to the reviewer's suggestion.

For the mouse tumor model, as shown in Fig R8, α PD-1-IL-12m effectively inhibited MC38 tumor growth as equimolar wild-type IL-12 did.

For humanized tumor model, as shown in Fig R9, wild-type human IL-12 showed robust tumor growth inhibition but also caused huge weight loss, and all mice did not survive over days 15 after treatment. In contrast, α hPD-1-hIL-12m can effectively inhibit tumor growth without weight loss.

Fig R8. C57BL/6 mice were inoculated with 5×10^5 MC38 tumor cells. Tumor-bearing mice ($n = 5$ /group) were intraperitoneally treated with PBS or $20 \mu\text{g}$ α PD-1-IL-12m or equimolar WT IL-12 on days 15, 18, and 21. The tumor growth of mice was recorded.

Fig R9. Humanized mice were inoculated with 2×10^6 A549 tumor cells. Tumor-bearing mice ($n = 4/\text{group}$) were intraperitoneally treated with PBS or equimolar wild-type hIL-12 or $\alpha\text{hPD-1-hIL-12m}$ on days 10, 13, 16, and 19. The tumor growth (A) and body weight (B) of mice were measured.

11) In the MC38-EGFR5 model, what is the relative expression of mouse EGFR versus mPD-1 on tumor cells?

Thanks to the reviewer's comment. In the MC38-EGFR5 tumor model, mouse EGFR5 was engineered to express on MC38 tumor cells, while PD-1 is mainly expressed on T cells but not tumor cells in TME. MC38-EGFR5 tumor model was used to compare the antitumor efficacies of tumor cell-targeting $\alpha\text{EGFR-IL-12m}$ and T cell-targeting $\alpha\text{PD-1-IL-12m}$. To confirm whether these fusion proteins preferentially bind to different cells, the cell suspension of MC38-EGFR5 tumor tissues is incubated with these fusion proteins. As shown in Fig R10, $\alpha\text{EGFR-IL-12m}$ mainly binds to tumor cells as a tumor cell-targeting fusion protein due to EGFR on tumor cells, while $\alpha\text{PD-1-IL-12m}$ prefers to bind to $\text{CD8}^+\text{T}$ cells as a T cell-targeting fusion protein. Therefore, we believe the $\alpha\text{PD-1}$ -mediated IL-12 *cis*-binding resulted in the restored bioactivity of IL-12 to activate $\text{CD8}^+\text{T}$ cells in tumors and, consequently, more robust antitumor immunity.

Fig R10. Digested tumor tissues from the MC38-EGFR5 tumor were incubated with $\alpha\text{EGFR-IL-12m}$ or $\alpha\text{PD-1-IL-12m}$. Protein binding to tumor cells or $\text{CD8}^+\text{T}$ cells was detected by flow cytometric analysis.

12) How effective is WT IL-12 at 5 ug doses? Comparison between the efficacy of $\alpha\text{PD-1-mIL-12}$ and $\alpha\text{PD-1-IL-12}$ in tumor models side by side with toxicity data is needed.

Thanks for the review's comment. As shown in Fig R11, we compared the antitumor efficacy and toxicity of wild-type IL-12, α PD-1-IL-12wt, and α PD-1-IL-12m side by side in the MC38 tumor model. IL-12 (green line) showed effective tumor control but significantly elevated inflammatory cytokines in serum compared to equimolar α PD-1-IL-12m, although the weight loss was not evident. The fusion protein α PD-1-IL-12wt (blue line) also showed a robust antitumor effect. However, it was accompanied by more severe side effects, including huge weight loss and higher levels of inflammatory cytokines in serum. The fusion protein α PD-1-IL-12m (red line) showed the same antitumor effect as IL-12 or α PD-1-IL-12wt did. More importantly, α PD-1-IL-12m was well tolerated and barely caused any side effects, neither the weight loss nor the increased inflammatory cytokines. Collectively, compared to IL-12 or α PD-1-IL-12wt, α PD-1-IL-12m can induce robust antitumor responses with minimal adverse effects.

Fig R11. C57BL/6 mice were inoculated with 5×10^5 MC38 tumor cells. Tumor-bearing mice ($n = 5/\text{group}$) were intraperitoneally treated with PBS or $5 \mu\text{g}$ α PD-1-IL-12m or equimolar wild-type IL-12 or α PD-1-IL-12wt on days 15, 18, and 21. (A) The tumor growth of mice was recorded. (B) The body weight change curve of mice. (C) After the final treatment, inflammation cytokines in serum were measured. The P value was determined by two-way ANOVA (A, B) or one-way (C). * $P < 0.05$, ** $P < 0.01$, *** $P < 0.001$, **** $P < 0.0001$.

Reviewer #3 (expert in biomedical engineering and IL-12-based therapies):

This manuscript describes the development of a novel immunocytokine comprised of anti-PD-1 fused to a low-affinity IL-12 construct. The goal of the study was to target IL-12, whose clinical development has been stymied by serious toxicities, to TILs via anti-PD-1 while minimizing signaling in peripheral lymphocytes, particularly NK cells, via the mutated IL-12. Impressively, systemic administration of the immunocytokine did not result in toxicities that are commonly associated with IL-12. Furthermore, studies demonstrate that anti-PD-1 engagement resulted in cis-binding and subsequent signaling of low affinity IL-12.

Although the authors claim an abscopal effect, this is debatable since an intratumoral injection results in systemic uptake.

Thanks to the reviewer's comment. We used FTY720 to dissect this confusing condition in Figure 6. FTY720 is a small-molecule analog of sphingosine 1-phosphate (S1P). FTY720 treatment induces the internalization and degradation of the S1P receptor, thus preventing lymphocyte egress from the LNs⁸. We have confirmed that the systemic injection of α PD-1-IL-12m can effectively suppress tumor growth, which was not affected by the FTY720 blockade (Fig 5F). In the MC38 double-flank tumor model, if the fusion protein leaks from the injection site (right) and is taken up by the distal tumor (left), the suppression of distal tumor growth should be observed, which should not be affected by the FTY720 blockade. However, we observed that the FTY720 blockade completely abolished the antitumor effect of α PD-1-IL-12m on the distal (left) tumor (Fig 6C), suggesting that the fusion protein leakage is irrelevant or is not sufficient to impact the abscopal effect. It is more likely that the tumor-specific CD8⁺T cells activated by α PD-1-IL-12m in the primary (right) tumor contributed to the systemic antitumor immune responses to control distal (left) tumor via lymph and blood circulation.

This strategy requires TILs expressing PD-1 and would not appear to be effective in cold, excluded tumors. This caveat should be added.

Another limitation of the manuscript is that it does not address immune related toxicities of anti-PD-1 which are not often reproduced in mice. The fusion of IL-12 is likely to exacerbate autoimmune related toxicities in humans.

Thanks to the reviewer's suggestion. The revised manuscript discusses the limitations of α PD-1-IL-12m for tumor therapy.

Although this novel T cell-targeting IL-12 fusion protein can preferentially activate tumor-infiltrating CD8⁺T cells and induce robust antitumor immune responses with limited side effects, this protein may have several potential limitations. Firstly, as tumor-infiltrating CD8⁺T cells were identified to be primarily responsible for the antitumor effect of α PD-1-IL-12m (Fig 5B), the tumors with plenty of T cell infiltration, known as "hot" tumors, might well respond to α PD-1-IL-12m treatment. However, "cold" or immune-excluded tumors lacking enough T cell infiltration may resist the α PD-1-IL-12m treatment. Moreover, the PD-1 expression on CD8⁺T cells is also essential for the treatment efficacy of α PD-1-IL-12m as both the PD-1 targeting effect

and α PD-1-mediated *cis*-binding lead to the restoration of IL-12 stimulatory signaling to CD8⁺T cells. Therefore, additional immunotherapies⁹ to promote CD8⁺T cell activation¹⁰ and infiltration¹¹ in combination with α PD-1-IL-12m treatment hold the key to breaking tumor resistance to successfully treating more types of cancer.

Another limitation is the incomplete investigations of the immune-related adverse effects (irAEs) of α PD-1 therapy. PD-1/PD-L1 blockade releases the suppression and reactivates the immune system, leading to the development of irAEs, which may affect and even cause damage to multiple tissues and organs¹². The irAEs of α PD-1 therapy commonly occur in clinical cancer patients but do not often happen in mouse tumor models. Although this study observed no severe irAEs after α PD-1-IL-12m treatment, including serum cytokine, transaminase, and weight change (Fig 4E, 4F, 4G, S3G, and S3H), awareness should be raised, and more comprehensive investigations are urgently needed, especially some tumor-bearing hosts might have more PD-1 expression outside TME.

In addition, the timing of the body weight loss and toxicity is curious. Systemic IL 12 treatments typically result in a spike in IFN γ within 12-36 hrs which is associated with weight loss and elevated liver enzymes. It is not clear why toxicity is delayed until day 7-9 after treatment.

Although the detailed mechanism of how IL-12 and its downstream IFN- γ cause these side effects is not fully understood, studies from us and others suggest that the abnormal physical condition was progressively worsened, leading to severe weight loss during IL-12 treatment. Collectively, in terms of the timing points, these side effects showed a sequential order: a rapid increase in IFN- γ initially (Fig 4F), followed by an increase in transaminases (Fig 4G, S4H), and finally, a decrease in body weight (Fig 4E). In addition, fusing with anti-PD-1 antibodies may prolong the half-life of IL-12, resulting in delayed toxicity.

After IL-12 systemic administration, IL-12 results in a rapid increase in serum IFN- γ that is apparent as early as day one and peaks about day 3¹³. NK cells are primarily responsible for the rapidly elevated IFN- γ in serum, as the depletion of NK cells completely abolished this effect¹⁴. The elevated IFN- γ was the primary mediator of IL-12-caused side effects. IFN- γ can activate Kupffer cells and then cause hepatocytic injury with elevated levels of aminotransferases, including ALT and AST in serum¹⁵. In addition, IL-12 caused severe gastrointestinal toxicity with diarrhea, and substantial weight loss was substantially mitigated when IFN- γ signaling was blocked¹³. The body weight of mice gradually decreases and commonly reaches the lowest point at day 7 after treatment¹³. Moreover, IL-12 administration also caused other side effects such as splenomegaly, neutropenia, and lymphopenia¹⁶.

A number of additional minor concerns should be addressed:

Line 60 – the statement that ‘numerous strategies have been proposed to reduce side effects by local delivery’ should include a reference.

Thanks to the reviewer’s suggestion. Three references are cited here and added in the

revised manuscript.

#1 Wang P, Li X, Wang J, Gao D, Li Y, Li H, Chu Y, Zhang Z, Liu H, Jiang G, Cheng Z, Wang S, Dong J, Feng B, Chard LS, Lemoine NR, Wang Y. Re-designing Interleukin-12 to enhance its safety and potential as an anti-tumor immunotherapeutic agent. *Nat Commun.* 2017 Nov 9;8(1):1395. Doi: 10.1038/s41467-017-01385-8.

In this study, tumor-targeted oncolytic adenovirus (Ad-TD) producing non-secreting (ns) IL-12 was intraperitoneally injected to promote the retention of IL-12 locally in tumors and reduce the side effects caused by systemic exposure of IL-12.

#2 Hwang MP, Fecek RJ, Qin T, Storkus WJ, Wang Y. Single injection of IL-12 coacervate as an effective therapy against B16-F10 melanoma in mice. *J Control Release.* 2020 Feb; 318:270-278. Doi: 10.1016/j.jconrel.2019.12.035.

In this study, IL-12-encapsulated complex coacervate was developed to sustainably deliver IL-12 into the tumor microenvironment and improve the bioactivity of IL-12.

#3 Agarwal Y, Milling LE, Chang JYH, Santollani L, Sheen A, Lutz EA, Tabet A, Stinson J, Ni K, Rodrigues KA, Moyer TJ, Melo MB, Irvine DJ, Wittrup KD. Intratumorally injected alum-tethered cytokines elicit potent and safer local and systemic anticancer immunity. *Nat Biomed Eng.* 2022 Feb;6(2):129-143. Doi: 10.1038/s41551-021-00831-9.

In this study, intratumorally injected alum-tethered IL-12 increased the retention of cytokines in tumor tissue, eliminated systemic toxicities, and achieved increased antitumor efficacy.

Line 72 – ‘PD-1 has much higher expression inside TME...’ should include a reference. Thanks to the reviewer’s suggestion. Two references are cited here and added in the revised manuscript.

#1 Gros A, Robbins PF, Yao X, Li YF, Turcotte S, Tran E, Wunderlich JR, Mixon A, Farid S, Dudley ME, Hanada K, Almeida JR, Darko S, Douek DC, Yang JC, Rosenberg SA. PD-1 identifies the patient-specific CD8⁺ tumor-reactive repertoire infiltrating human tumors. *J Clin Invest.* 2014 May;124(5):2246-59. Doi: 10.1172/JCI73639.

In this study, the phenotypes of CD8⁺TILs from human melanomas were characterized, and it was found that PD-1 was highly expressed on CD8⁺TILs and used to identify the clonally expanded tumor-reactive lymphocytes.

#2 Shen J, Zou Z, Guo J, Cai Y, Xue D, Liang Y, Wang W, Peng H, Fu YX. An engineered concealed IL-15-R elicits tumor-specific CD8⁺T cell responses through PD-1-cis delivery. *J Exp Med.* 2022 Dec 5;219(12): e20220745. Doi: 10.1084/jem.20220745.

In this study, the PD-1 expression on T cells from several tissues of tumor-bearing mice was detected, and the result showed that T cells in tumor tissue exclusively expressed a much higher level of PD-1 while barely expressed PD-1 in other tissues.

Line 80 – T cells are not ‘naïve’; only CD4 and CD8 markers were used.
We have replaced “naïve” with “splenic” to define these cells in the revised manuscript.

Line 92 – the disposability of NK cells is model dependent; authors should not claim that NK cells are not important in IL-12 immunotherapy.

We agree with the reviewer. The role of NK cells in IL-12 immunotherapy is indeed model-dependent. In the subcutaneous MC38 tumor model, NK cells are dispensable for the antitumor efficacy of IL-12. We cannot exclude the essential role of NK cells in IL-12 based immunotherapies for other tumor models. We have added this point of view in the revised manuscript.

Line 136 – no detail about anti-PD-1, either the mouse or human version
We have clarified the origin of anti-PD-1 antibodies used in this study in the method section:

Mouse anti-PD-1 antibody clone: J43.

Human anti-PD-1 antibody clone: Pembrolizumab (Keytruda).

Line 137 – mIL-12 is the standard nomenclature for murine IL-12. It is strongly suggested that the authors do not use mIL-12 to mean mutated IL-12. This is very confusing.

Thanks to the reviewer’s suggestion. The fusion proteins are renamed as follows for a more accurate description in the revised manuscript.

Table 1. Abbreviation for fusion proteins

Fusion protein forms	Abbreviation
mouse IL-12 wild type	IL-12
anti-PD-1-mouse IL-12 wild type	α PD-1-IL-12wt
anti-PD-1-mouse IL-12 Mut-1	α PD-1-IL-12m1
anti-PD-1-mouse IL-12 Mut-2	α PD-1-IL-12m
human IL-12 wild type	hIL-12
anti-PD-1-human IL-12 wild type	α PD-1-hIL-12wt
anti-PD-1-human IL-12 Mut-1	α PD-1-hIL-12m1
anti-PD-1-human IL-12 Mut-2	α PD-1-hIL-12m
anti-human PD-1-human IL-12 Mut-2	α hPD-1-hIL-12m

Line 150 – the statement that the concentration of α PD-1-mIL-12 was obviously higher in the tumor is not correct. Fig S2D shows similar concentrations in the liver and spleen. Thanks to the reviewer’s comment. We repeated this experiment to track the tissue distribution of α PD-1-IL-12m in different conditions. We treated mice with 20ug α PD-1-IL-12m, the effective dose in this tumor model, and collected mouse organs and tissues 24 hours later. As shown in Fig R1, the concentration of α PD-1-IL-12m in the tumor is significantly higher than in other tissues, including the liver and spleen. This

result confirms that α PD-1-IL-12m is preferentially accumulated in tumors. We have added this result in the revised manuscript.

Fig R1. MC38 tumor-bearing mice (n = 5) were i.p. injected with 20 μ g α PD-1-IL-12m and sacrificed 24h after treatment. Mice tissues were collected and homogenized. The concentration of fusion protein in the supernatant was measured and normalized by ELISA. The P value was determined by one-way. *P < 0.05, **P < 0.01, ***P < 0.001, ****P < 0.0001.

Line 152 – data are not convincing that α PD-1-mIL-12 ‘preferentially accumulates in tumors.

Thanks to the reviewer’s comment. As shown in Fig R1 in the last comment, the concentration of α PD-1-IL-12m in tumors is significantly higher than in other tissues, including the liver and kidney. This result confirms that α PD-1-IL-12m is preferentially accumulated in tumors. We have added this result in the revised manuscript.

Line 175 – why doesn’t α PD-1-mIL-12 result in higher signaling than α EGFR-mIL-12 in HEK-mPD-1 cells (Fig S3C)?

In Fig S3C, the IL-12 mutant form in α PD-1-IL-12m1 was Mut-1. Different from Mut-2, the binding affinity of Mut-1 is the same as wild-type IL-12. The addition of α PD-1 antibody had less effect on its binding affinity to its receptors. Therefore, α PD-1-IL-12m1 did not result in higher signaling than α EGFR-IL-12m1 in HEK-mPD-1 cells (Fig S3C).

Mut-2’s binding affinity to the IL-12 receptor was severely impaired (Fig 2C, S2D). Thus, we utilized α PD-1-mediated *cis*-binding to facilitate the binding of Mut-2 to its receptors, consequently resulting in restored signaling in HEK-mPD-1 cells (Fig 3A, 3B).

Line 202 – mice also treated on days 17 and 20 (not just day 14)

Thanks to the reviewer for pointing out the misdescription. It has been corrected as “The well-established tumors were treated beginning on days 14” in the revised manuscript.

Line 258 – Fig S4F does not appear to demonstrate that FTY720 diminishes T cells in the periphery.

Thanks to the reviewer’s comment. To further confirm the blockade efficacy of FTY720 in our experiments, we continuously monitored the frequency of T cells in peripheral blood. As shown in Fig R12, the absolute number and the frequency of CD3⁺T cells decreased rapidly in the first 6 hours after FTY720 treatment and maintained a shallow level until 48 hours. In our experiments, FTY720 was used once every other day. Therefore, we confirmed that FTY720 treatment can diminish T cells in the periphery during the experimental period.

Fig R12. Tumor-bearing mice were intraperitoneally injected with 25μg FTY720, and the blood samples were collected at different time points after FTY720 treatment. The absolute number (A) and the frequency of CD3⁺T cell (B) in blood were detected.

Line 281 – intratumoral injections, like s.c. injections result in systemic distribution
As discussed above, in the MC38 double-flank tumor model, if the fusion protein leaks from the injection site (right) and is taken up by the distal tumor (left), the suppression of distal tumor growth should be observed, which should not be affected by FTY720 blockade. However, we observed that the FTY720 blockade completely abolished the antitumor effect of αPD-1-IL-12m on the distal (left) tumor (Fig 6C), suggesting that the fusion protein leakage is irrelevant or is not sufficient to impact the abscopal effect. The tumor-specific CD8⁺T cells activated by αPD-1-IL-12m in the primary (right) tumor contributed to the systemic antitumor immune responses to control distal (left) tumor via lymph and blood circulation.

Line 572 – for all graphs reporting ‘Days after treatment’; please clarify is this after 1, 2,3 etc injections?

“Days after treatment” in this study means “Days after the first treatment,” and we have

clarified it in the revised manuscript.

Line 601 – the ‘Ctrl’ group in Fig 3C is not defined.

In Fig 3C, the “Ctrl” group means that the cell culture medium without fusion protein was used to stimulate HEK-mPD-1 cells as a negative control, and we have clarified it in the revised manuscript.

Line 713 – when after treatment were tumors harvested?

The tumors were harvested 48 hours after the second treatment and processed for further analysis, and we have clarified it in the revised manuscript.

References:

1. Sakuishi K, Apetoh L, Sullivan JM, Blazar BR, Kuchroo VK, Anderson AC. Targeting Tim-3 and PD-1 pathways to reverse T cell exhaustion and restore anti-tumor immunity. *J Exp Med* **207**, 2187-2194 (2010).
2. Xue D, *et al.* A tumor-specific pro-IL-12 activates preexisting cytotoxic T cells to control established tumors. *Sci Immunol* **7**, eabi6899 (2022).
3. McLane LM, Abdel-Hakeem MS, Wherry EJ. CD8 T Cell Exhaustion During Chronic Viral Infection and Cancer. *Annu Rev Immunol* **37**, 457-495 (2019).
4. Bae J, *et al.* IL-2 delivery by engineered mesenchymal stem cells re-invigorates CD8(+) T cells to overcome immunotherapy resistance in cancer. *Nat Cell Biol* **24**, 1754-1765 (2022).
5. Fourcade J, *et al.* Upregulation of Tim-3 and PD-1 expression is associated with tumor antigen-specific CD8+ T cell dysfunction in melanoma patients. *J Exp Med* **207**, 2175-2186 (2010).
6. Pauken KE, *et al.* Epigenetic stability of exhausted T cells limits durability of reinvigoration by PD-1 blockade. *Science* **354**, 1160-1165 (2016).
7. Fang Y, Malik M, England SK, Imoukhuede PI. Absolute Quantification of Plasma Membrane Receptors Via Quantitative Flow Cytometry. *Methods Mol Biol* **2475**, 61-77 (2022).
8. Chiba K. FTY720, a new class of immunomodulator, inhibits lymphocyte egress from secondary lymphoid tissues and thymus by agonistic activity at sphingosine 1-phosphate receptors. *Pharmacol Ther* **108**, 308-319 (2005).
9. Galon J, Bruni D. Approaches to treat immune hot, altered and cold tumours with combination immunotherapies. *Nat Rev Drug Discov* **18**, 197-218 (2019).
10. Horton BL, *et al.* Lack of CD8(+) T cell effector differentiation during priming mediates checkpoint blockade resistance in non-small cell lung cancer. *Sci Immunol* **6**, eabi8800 (2021).
11. Tang H, *et al.* Facilitating T Cell Infiltration in Tumor Microenvironment Overcomes Resistance to PD-L1 Blockade. *Cancer Cell* **30**, 500 (2016).
12. Su C, *et al.* Adverse Effects of Anti-PD-1/PD-L1 Therapy in Non-small Cell Lung Cancer. *Front Oncol* **10**, 554313 (2020).
13. Leonard JP, *et al.* Effects of single-dose interleukin-12 exposure on interleukin-12-associated toxicity and interferon-gamma production. *Blood* **90**, 2541-2548

- (1997).
14. Mansurov A, *et al.* Masking the immunotoxicity of interleukin-12 by fusing it with a domain of its receptor via a tumour-protease-cleavable linker. *Nat Biomed Eng* **6**, 819-829 (2022).
 15. Car BD, *et al.* Role of interferon-gamma in interleukin 12-induced pathology in mice. *Am J Pathol* **147**, 1693-1707 (1995).
 16. Lenzi R, *et al.* Phase I study of intraperitoneal recombinant human interleukin 12 in patients with Müllerian carcinoma, gastrointestinal primary malignancies, and mesothelioma. *Clin Cancer Res* **8**, 3686-3695 (2002).

REVIEWER COMMENTS

Reviewer #1 (expert in translational immunology and cancer therapy):

They've sufficiently addressed my comments. Recommend accept

Reviewer #1, commenting on the responses to Reviewer #3:

I have looked at the third review and the author's responses to the review.

In my opinion the authors responded to the review sufficiently and based on that, would recommend accept.

Reviewer #2 (expert in protein engineering and translational immunology):

I did not find that the authors addressed the feedback sufficiently for my comments, and many of the experiments performed were not responsive to the critiques and, in some cases, drew conclusions that were not supported by the data. The manuscript is still very difficult to read and understand. Further revisions are needed for this manuscript to be acceptable for publication.

- 1) The writing in the article still needs significant improvement, with particular attention to grammar, flow, and phrasing.
- 2) The response to point 1 does not make sense. The results in Figure R4 look very different from those presented in Figure 2C, yet the concentrations are indicated as being the same. Which graph is correct?
- 3) It is still not clearly indicated how the asymmetric dimer was designed.
- 4) This looks fine.
- 5) This is corrected.
- 6) The on-cell binding is helpful, but kinetics measurements on BLI or SPR to understand the molecular effects of IL-12 conjugation on PD-1 interaction. Moreover, the reduced affinity for PD-L1 of the antibody/IL-12 fusion compared to the unconjugated antibody should be mentioned in the text. However, the investigators did not respond to the reviewer request. The experiment that was performed does not speak to the relative contributions of PD-L1 and IL-12 binding since it is performed on a PD-1-expressing reporter HEK cell line, which does not reflect the surface levels of PD-1 or IL-12 receptor subunits on immune cells. The authors need to measure IL-12 cytokine/receptor binding both for the soluble cytokine and the antibody-fused molecule for comparison to PD-1/anti-PD-1 antibody binding energetics.
- 7) This is addressed.
- 8) Apologies for the confusion in my comment. The question was intended to ask for quantification of the number of PD-1, IL-12Rb1, and IL-12Rb2 receptors on immune cells in the tumor microenvironment, and how these compared to those on the reporter cell line. No quantification (number of receptors per cell) was provided, as requested. Also, why isn't PD-1 quantification on the intratumoral CD8+ T cells presented?
- 9) This is addressed.
- 10) This is addressed.

11) Again, apologies for the confusion. The question was intended to request comparison of EGFR expression on tumor cells to PD-1 expression in the tumor microenvironment. Quantification (number of receptors per cell) should be determined.

Reviewer #3 (expert in biomedical engineering and IL-12-based therapies):

Absent - the responses to this reviewer were assessed by Reviewer #1.

RESPONSE TO REVIEWERS' COMMENTS

Reviewer #1 (expert in translational immunology and cancer therapy):

They've sufficiently addressed my comments. Recommend accept

Reviewer #1, commenting on the responses to Reviewer #3:
I have looked at the third review and the author's responses to the review.

In my opinion the authors responded to the review sufficiently and based on that, would recommend accept.

Reviewer #2 (expert in protein engineering and translational immunology):

I did not find that the authors addressed the feedback sufficiently for my comments, and many of the experiments performed were not responsive to the critiques and, in some cases, drew conclusions that were not supported by the data. The manuscript is still very difficult to read and understand. Further revisions are needed for this manuscript to be acceptable for publication.

1) The writing in the article still needs significant improvement, with particular attention to grammar, flow, and phrasing.

Thanks for the reviewer's suggestion. The manuscript has been finely revised and polished by Springer Nature Author Services.

2) The response to point 1 does not make sense. The results in Figure R4 look very different from those presented in Figure 2C, yet the concentrations are indicated as being the same. Which graph is correct?

For the reviewer's convenience, we have appended a list of relevant figure panels from both the revised manuscript and the first response to this correspondence.

The distinctive binding curves were derived from experiments utilizing two distinct cell lineages: in vitro activated T cells and reporter cells. In an effort to achieve strong and saturating binding at reasonable IL-12-Fc fusion protein concentrations, we switched to utilize reporter cells in on-cell titrations instead of immune cells that present faint fusion protein binding in vitro. Despite employing identical fusion protein concentrations in both experiments, the observed surrogate binding capacity, as measured by mean fluorescence intensities (MFIs), appeared to be lower in spleen T cells (Fig.2C) than in reporter cells (Fig.R4), implying differential IL-12 receptor expression levels between the two cell types. Analysis of the expression levels and species of IL-12 receptors between these two types of cells was conducted (refer to Fig.RR4-5).

Notably, although reporter cells express human IL-12R, mouse IL-12 can still bind to human IL-12R, thereby eliciting IL-12 signaling within these cells. Furthermore, Fig.R4 aligns with Fig.2C, illustrating that Mut-2's binding affinity to the receptor is markedly impaired, whereas Mut-1's remains comparable to the wild type (WT). We may replace Fig. 2C with Fig.R4, which presents improved curves for on-cell binding of wild type and mutant IL-12 to IL-12 receptors.

3) It is still not clearly indicated how the asymmetric dimer was designed.

We appreciate the reviewer's kind feedback. The construction of this asymmetric dimer (illustrated in Fig.RR1) employs Knobs-into-Holes technology¹. Specific mutations within the Fc CH3 domains facilitate the accurate pairing of two heavy chains, with only one heavy chain linked to IL-12 at the C-terminus (Fig 2D).

Fig RR1. Knob-into-hole CH3 mutations ensure heterodimer not homodimer formation¹.

4) This looks fine.

5) This is corrected.

6) The on-cell binding is helpful, but kinetics measurements on BLI or SPR to understand the molecular effects of IL-12 conjugation on PD-1 interaction. Moreover, the reduced affinity for PD-L1 of the antibody/IL-12 fusion compared to the unconjugated antibody should be mentioned in the text. However, the investigators did not respond to the reviewer request. The experiment that was performed does not speak to the relative contributions of PD-L1 and IL-12 binding since it is performed on a PD-1-expressing reporter HEK cell line, which does not reflect the surface levels of PD-1 or IL-12 receptor subunits on immune cells. The authors need to measure IL-12 cytokine/receptor binding both for the soluble cytokine and the antibody-fused molecule for comparison to PD-1/anti-PD-1 antibody binding energetics.

Thanks for the reviewer's comment. In response to the reviewer's inquiry, we have assessed the contribution of the anti-PD-1 moiety within the fusion protein to binding PD-1-IL-12m to T cells.

In this experiment, to precisely assess the impacts of anti-PD-1 and IL-12m on the fusion protein's interaction with T cells, we employed α EGFR-IL-12m as a control molecule. This control protein contains identical monovalent IL-12m and a similar configuration surrounding IL-12 within the Ab-IL-12m-associated fusion proteins. The anti-Fc antibody was utilized to quantify the fusion protein's binding to immune cells.

As illustrated in Figure RR2, when tested on in vitro activated T cells, α PD-1-IL-12m (red line) exhibits a more robust binding capacity compared to α EGFR-IL-12m (green line). Additionally, α PD-1-IL-12m (red line) demonstrates an unaltered affinity for immune cells compared to the unconjugated antibody, contrary to the observations with reporter cells (Fig. R5 in the initial response). Given the significant variances in PD-1 and IL-12 receptor expression levels between immune cells and reporter cells (as presented in Fig RR4-5), it remains plausible that the influence of IL-12 on α PD-1's binding affinity for PD-1 could be cell-dependent. Nonetheless, the pivotal role of the α PD-1 antibody moiety in enhancing the fusion protein's

binding affinity for immune cells is further substantiated, aligning with the observations presented in Fig.R5 of the initial response.

Fig RR2. *In vitro* activated T cells were incubated with serially diluted α EGFR-IL-12m, α PD-1-IL-12m, or anti-PD-1 antibody. Protein binding to T cells was detected by flow cytometric analysis. MFI: mean fluorescence intensity.

7) This is addressed.

8) Apologies for the confusion in my comment. The question was intended to ask for quantification of the number of PD-1, IL-12Rb1, and IL-12Rb2 receptors on immune cells in the tumor microenvironment, and how these compared to those on the reporter cell line. No quantification (number of receptors per cell) was provided, as requested. Also, why isn't PD-1 quantification on the intratumoral CD8+ T cells presented?

We appreciate the reviewer's comment and regret the delay in our response. Quantitative Flow Cytometry enables the determination of absolute receptor numbers on the plasma membrane². However, obtaining the necessary assay kits, validated antibodies, and reagents proved to be time-consuming. Finally, we have been able to get two special antibodies to explore receptor quantification.

To establish a standard curve (Fig RR3, Table RR1), we followed the protocol using the Quantibrite™ PE Phycoerythrin Fluorescence Quantitation Kit (BD, 340495), which employs beads conjugated with four standard levels of PE fluorescence. Receptor quantification per cell was achieved using PE-

conjugated antibodies specific to the receptors of interest. Additionally, we tested with a series of titration of antibody concentrations to ensure saturated binding to the target cell membrane receptors.

Through this methodology, we obtained the counts of PD-1 and IL-12R on intratumoral CD8⁺T cells and reporter cells. The PD-1 count approximates 18,000 per CD8⁺T cell within tumors (Fig RR5A), significantly lower than the count of 500,000 per cell seen on reporter cells (Fig RR4D), which were artificially transfected with PD-1. The IL-12Rβ2 count is approximately 4,000 per CD8⁺T cell within tumors (Fig RR5B), while the expression of IL-12Rβ1 was not detectable using the available PE-labeled IL-12Rβ1 antibody (Fig RR5C). Similarly, the IL-12Rβ1 count is approximately 45,000 per reporter cell (Fig RR4E), while IL-12Rβ2 expression was not detected on reporter cells using the same PE-labeled IL-12Rβ2 antibody (Fig RR4C). Notably, while reporter cells express human IL-12R, mouse IL-12 can bind to human IL-12R and activate IL-12 signaling in these cells.

In summary, reporter cells exhibited significantly higher expression levels of PD-1 and IL12 receptors than intratumoral CD8⁺T cells, but the ratios of PD1 to IL12 receptor were relatively similar between these cells.

Fig RR3. 10000 PE beads were detected by flow cytometry at a specific voltage, which was applied for subsequent marker quantification. Distinctly separated four-peak signals of PE beads were illustrated on a PE spectrum (A). The four peaks corresponded to four beads carrying varying numbers of PE and were categorized as Low, Med Low, Med High, and High. The standard curve was derived through linear regression of Log₁₀ PE/Bead and

Log₁₀ Geo Mean (B), following the equation: $Y = 0.9993 * X - 0.2081$, where Y represents Log₁₀ Geo Mean and X represents Log₁₀ PE/Bead.

Marker	Events	Geometric Means	Lg Geometric Means	PE Molecules/Bead	Lg PE Molecules/Bead
All	9407	7146	Y value		X value
Low	2265	621	2.793	1000	3
Med Low	2290	7691	3.886	12445	4.095
Med High	2410	15602	4.193	26427	4.422
High	2379	47309	4.675	74906	4.875

Table RR1. Statistics of Geometric Mean and PE molecule quantities of the four PE beads. Geometric Mean was assessed by flow cytometry, and data on PE Molecules per Bead was gathered following the kit instructions. Log₁₀ Geometric Mean and Log₁₀ PE Molecules/Bead were calculated and plotted for the standard curve in Fig RR3B.

D

E

Fig RR4. The Geo Means of PE-labeled anti-PD1 (A), anti-IL12Rβ1 (B) and anti-IL12Rβ2 (C) binding to the HEK-IL12-mPD1 cell line were identified using flow cytometry with a series titration of PE-antibody. The Geo Means obtained from A and B were substituted into the equation to calculate mean PD1 and IL12Rβ1 counts per cell. The PE-anti-PD1 and PE-anti-IL12Rβ1 antibody were saturated at 8 μL (D) and 15 μL (E) respectively within a non-linear fitting curve with increasing titration of the PE antibody.

A

B

C

Fig RR5. C57BL/6 mice were inoculated with 5×10^5 MC38 tumor cells. On day 14, MC38 tumors were collected and processed. The Geo Means of PE-

labeled anti-PD1, anti-IL12R β 1 (C) and anti-IL12R β 2 binding to the intratumoral CD8⁺T cells were identified via flow cytometry. The mean count of PD1 (A) or IL12R β 2 (B) on CD8⁺T cells was quantified at the saturation concentration according to the fitting curve in FigRR4. Data are presented as mean \pm SD from two independent experiments.

9) This is addressed.

10) This is addressed.

11) Again, apologies for the confusion. The question was intended to request comparison of EGFR expression on tumor cells to PD-1 expression in the tumor microenvironment. Quantification (number of receptors per cell) should be determined.

Thanks for the reviewer's comment. As illustrated in the Figure RR5A, the count of PD-1 approximates 18,000 per CD8⁺T cell within tumors. Utilizing the same method, we also determined the number of EGFR to be almost 45,000 per tumor cell (Fig RR6B), significantly higher than the count of PD-1 on immune cells. Furthermore, to directly assess the efficacy of these fusion proteins in binding to their target cells, we incubated tumor tissue suspensions with separate fusion proteins. As illustrated in Fig RR7, α PD-1-IL-12m predominantly binds to CD8⁺T cells rather than tumor cells, whereas α EGFR-IL-12m primarily binds to tumor cells rather than CD8⁺T cells. Most notably, the binding kinetics of α EGFR-IL-12m with tumor cells are even more potent than those of α PD-1-IL-12m with CD8⁺T cells. This outcome demonstrates the potent binding capability of α EGFR-IL-12m to tumor cells. However, the tumor-binding of α EGFR-IL-12m was not able to elicit IL-12 bioactivity in T cells through a *cis*-binding mechanism, as observed with α PD-1-IL-12m, resulting in a weaker antitumor effect.

Fig RR6. The Geo Means of PE-labeled anti-EGFR binding to the MC38-EGFR5 cell line were identified using flow cytometry with a series of PE-antibody titrations (A). The Geo Means obtained from A were substituted into the equation to determine the mean EGFR counts/cell. The PE-anti-EGFR antibody was saturated at 8 μ L within a non-linear fitting curve with increasing titration of PE antibody (B).

Fig RR7. Digested tumor tissues from the MC38-EGFR5 tumors were incubated with α EGFR-IL-12m or α PD-1-IL-12m. Fusion protein binding to the tumor cells or CD8⁺T cells was detected by flow cytometric analysis.

1. Merchant AM, *et al.* An efficient route to human bispecific IgG. *Nat Biotechnol* **16**, 677-681 (1998).
2. Fang Y, Malik M, England SK, Imoukhuede PI. Absolute Quantification of Plasma Membrane Receptors Via Quantitative Flow Cytometry. *Methods Mol Biol* **2475**, 61-77 (2022).

Appendix:

Related results in the manuscript and the first response:

Fig 2C. *In vitro* activated T cells were incubated with WT IL-12, Mut-1, or Mut-2. Protein binding to T cells was detected by flow cytometric analysis. MFI: mean fluorescence intensity.

Note: Every IL-12 fusion protein in this panel is an IL-12-Fc version without anti-PD-1 Fab.

Fig R4. HEK cells were incubated with serially diluted WT IL-12, Mut-1, or Mut-2. Protein binding to HEK cells was detected by flow cytometric analysis. MFI: mean fluorescence intensity.

Note: Every IL-12 fusion protein in this panel is an IL-12-Fc version without anti-PD-1 Fab.

Fig R5. HEK-mPD-1 cells were incubated with serially diluted hIgG1, αEGFR-IL-12m, αPD-1-IL-12m, or anti-PD-1 antibody *in vitro*. Protein binding to HEK-mPD-1 cells was detected by flow cytometric analysis.

REVIEWERS' COMMENTS

Reviewer #2 (Remarks to the Author):

Although the authors did a much better job of addressing the points that were raised, they did not add most of the important new data back into the manuscript. In particular, the receptor quantification data should be presented and discussed in the text.

RESPONSE TO REVIEWERS' COMMENTS

Reviewer #2 (Remarks to the Author):

Although the authors did a much better job of addressing the points that were raised, they did not add most of the important new data back into the manuscript. In particular, the receptor quantification data should be presented and discussed in the text.

Thanks for the reviewer's comment. We have added the receptor quantification results (Fig RR3-6) in the revised manuscript supplementary Fig. 5 and discussed in the text.